# The Female Pelvic Floor Fascia Anatomy: A Systematic Search and Review

**DOI:** 10.3390/life11090900

**Published:** 2021-08-30

**Authors:** Mélanie Roch, Nathaly Gaudreault, Marie-Pierre Cyr, Gabriel Venne, Nathalie J. Bureau, Mélanie Morin

**Affiliations:** 1Research Center of the Centre Hospitalier Universitaire de Sherbrooke, Faculty of Medicine and Health Sciences, School of Rehabilitation, Université de Sherbrooke, Sherbrooke, QC J1H 5N4, Canada; melanie.roch@usherbrooke.ca (M.R.); nathaly.gaudreault@usherbrooke.ca (N.G.); marie-pierre.cyr@usherbrooke.ca (M.-P.C.); 2Anatomy and Cell Biology, Faculty of Medicine and Health Sciences, McGill University, Montreal, QC H3A 0C7, Canada; gabriel.venne@mcgill.ca; 3Centre Hospitalier de l’Université de Montréal, Department of Radiology, Radio-Oncology, Nuclear Medicine, Faculty of Medicine, Université de Montréal, Montreal, QC H3T 1J4, Canada; nathalie.bureau@umontreal.ca

**Keywords:** anatomy, connective tissue, fascia, pelvic floor, systematic review, women

## Abstract

The female pelvis is a complex anatomical region comprising the pelvic organs, muscles, neurovascular supplies, and fasciae. The anatomy of the pelvic floor and its fascial components are currently poorly described and misunderstood. This systematic search and review aimed to explore and summarize the current state of knowledge on the fascial anatomy of the pelvic floor in women. Methods: A systematic search was performed using Medline and Scopus databases. A synthesis of the findings with a critical appraisal was subsequently carried out. The risk of bias was assessed with the Anatomical Quality Assurance Tool. Results: A total of 39 articles, involving 1192 women, were included in the review. Although the perineal membrane, tendinous arch of pelvic fascia, pubourethral ligaments, rectovaginal fascia, and perineal body were the most frequently described structures, uncertainties were identified in micro- and macro-anatomy. The risk of bias was scored as low in 16 studies (41%), unclear in 3 studies (8%), and high in 20 studies (51%). Conclusions: This review provides the best available evidence on the female anatomy of the pelvic floor fasciae. Future studies should be conducted to clarify the discrepancies highlighted and accurately describe the pelvic floor fasciae.

## 1. Introduction

Up to 47% of women suffer from at least one pelvic floor disorder, including chronic pelvic pain, urinary incontinence, and pelvic organ prolapse [1,2]. With evidence to suggest that up to one-third of these women will have two or more pelvic floor disorders [1,3,4,5], one in five will require surgical intervention [6,7]. The significant burden of pelvic floor disorders is well documented, including both the impact on women’s quality of life and the associated personal and societal financial cost [8].

Scientific evidence has highlighted the importance of the musculature closing the base of the pelvis [9,10,11] and has suggested the potential involvement of the surrounding fasciae in the pathophysiology of pelvic floor disorders [12,13,14,15,16]. For instance, the pelvic floor fasciae are thought to contribute to urethral and pelvic organ support, thereby preventing incontinence and prolapse [13,14,15,16,17,18,19]. Accordingly, corrections made to the pelvic floor connective structures form the basis of modern surgical procedures for incontinence and prolapse [15,18,20,21,22,23,24]. The pelvic floor fasciae could also play a key role in chronic pelvic pain. Recent evidence suggests that the fasciae are involved in several chronic pain conditions, including low back pain and neck pain [25,26]. The structures that form the pelvic region are arranged as a series of muscles, fascia layers that are interwoven with vascular and nervous networks and connected to the various viscera. Moreover, the presence of nociceptors in the fasciae suggests that they may be a source of local and referred pain [27,28,29,30]. Structural and biomechanical alterations of the fasciae, such as thickening, reduced shear strain, and increased viscosity of hyaluronan, have also been suggested as pathophysiological mechanisms [25,31,32]. The very intimate relationship between the fasciae, muscles, and neurovascular structures also implies that these alterations in the fasciae could contribute to muscle tension and nerve sensitization [27].

Although there are numerous suspected mechanisms pertaining to the involvement of fasciae in incontinence, prolapse, and pain, scientific knowledge on this topic is still in its infancy. Moreover, there has been a worldwide proliferation of different forms of exercise as well as manual and alternative therapies targeting the fascial system. These therapies are used by clinicians as treatment options despite the fact that current evidence is insufficient to warrant their use [31]. A thorough understanding of the involvement of fasciae in the pathophysiology of pelvic floor disorders is needed but is currently limited due to the lack of consensus on their anatomy and related terminology [32,33,34]. Conflicting information is undeniably prominent with respect to female pelvic floor fasciae as they are interconnected, and organized in juxtaposed and nested layers oriented in different directions [34,35,36]. Moreover, various terms are often used to define the same structure [34,35,37]. However, there are also examples in the literature where the same term is used for more than one structure [19]. Even textbooks provide different descriptions for the same arrangement of connective tissue or fasciae, which leads to confusion and misunderstanding [38,39]. In addition, the structures are categorized using various terms such as fascia, membrane, septum, and ligament whereas the characteristics defining and discriminating each term remain unclear [37]. Gaps in knowledge of the anatomy of the pelvic floor fasciae have probably been the greatest obstacle in understanding the pathophysiology of pelvic floor disorders and hence the development of effective treatment avenues. The dual function assumed by these structures (i.e., flexibility and adaptability for childbirth and intercourse versus stiffness for continence and organ support) further complexify the physiology of the pelvic floor fasciae.

To shed light on the pathophysiology of pelvic floor disorders and to develop effective treatments, we must first understand the complexity of the fascial anatomy of the pelvic floor. Hence, the aim of this systematic search and review was to present the current state of knowledge on the macro- and micro-anatomy of the female pelvic fasciae.

## 2. Materials and Methods

### 2.1. Study Design

An exhaustive systematic search and review was performed to identify the literature relevant to the anatomy of the fasciae of the female pelvic floor. According to the typology of reviews proposed by Grant and Booth (2009) [40], this design was chosen because it typically addresses a broad-based question to provide a more accurate picture of the research on a given topic. It combines the strengths of a systematic review (a comprehensive search process) with those of a critical review, which presents, appraises, and synthesizes material from diverse sources [40]. Our systematic research and review was conducted in accordance with the proposed Checklist for Anatomical Reviews and Meta-Analyses (CARMA) [41] to ensure the reporting of evidence-based anatomy. Our work was also in line with the Preferred Reporting Items for Systematic Reviews and Meta-Analysis (PRISMA) statement [42]. The protocol was registered on the International Platform of Registered Systematic Review and Meta-analysis Protocol (identification number INPLASY202150067).

### 2.2. Search Strategy

A list of keywords was established a priori by a group of experts and by consulting traditional textbooks used in urology and gynecology [43,44,45,46]. The list was then cross-referenced with the Terminologica Anatomica [47] database to retrieve all the possible synonyms (Appendix A). The review specifically targeted the investing layer of fascia/connective tissue (i.e., fascia, ligament, membrane, septum, etc.) of the pelvic floor, which is a complex functional unit comprising structures located within the bony pelvis: the urogenital and anorectal viscera, the pelvic floor muscles, the fasciae/connective tissues, the nerves, and the blood vessels [48]. More specifically, the anatomical area of interest was delineated by the fasciae located between the pubic symphysis, anteriorly, to the sacrum/coccyx, posteriorly, and from the membranous layer of the perineal subcutaneous tissue, inferiorly, up to the endopelvic fascia, superiorly, which corresponds to the level of pelvic support II and III described by DeLancey [16]. The search strategy was developed by our team of experts and under the supervision of a professional librarian to identify all the tissue in the described region that can be considered fascia of the pelvic floor. The search was performed online, and targeted articles published in Medline (including AMED and CINAHL) and Scopus (including EMBASE and MEDLINE) and gray literature through the ProQuest database (for list of keywords, see Appendix B) from inception to 19 May 2021. Manual searches of previously published reviews and reference lists of the included articles were also conducted.

### 2.3. Inclusion and Exclusion Criteria

Observational studies such as case series, cohorts, case-control, and cross-sectional studies providing an anatomical description of the pelvic floor fasciae and connective tissue of women (cadaveric or living) without pelvic floor disorders or chronic pelvic pain were included. Studies were considered eligible if they included outcomes related to the description of macro- and/or micro-anatomy, using dissections, histological assessment, imaging, or clinical morphological measurements. Studies were excluded if they were not published in English, French, Italian, or Portuguese, and if the results presented combined women and men and/or fetuses and did not include a clear distinction between samples. Descriptions of pathological cases and animal studies were excluded, except when studies described a clear distinction of healthy adult human females. Editorial articles and surveys were also excluded. Systematic reviews were consulted to retrieve relevant references.

### 2.4. Study Selection and Data Collection Process

Two authors (M.R. and M.P.C.) independently reviewed the titles, abstracts, and full texts of articles in order to evaluate their eligibility. For all three phases, disagreements were resolved by consensus. If a disagreement could not be resolved, a third author (N.G. or M.M.) became involved. The percentage of agreement was used to assess the interrater agreement on the study’s eligibility. The level of agreement was described using an index proposed by Cicchetti [49] (i.e., percentage of agreement < 70% being considered poor; 70–79% fair; 80–89% good; and >90% excellent).

### 2.5. Data Extraction

The data were collected and extracted by two authors (M.R. and M.P.C.) using a standardized Excel file. The data extracted by each author were compared and combined, and a third author (N.G. or M.M.) was involved in cases of disagreement. The following information was extracted: author’s name, year of publication, study design, assessment approach, population characteristics (i.e., embalmed or fresh cadavers, or living females and age), and fascia-related outcomes (i.e., names of the structures, description and course, length, width, thickness, and histological features). The fascial system will be presented sequentially throughout the manuscript, starting with the most inferior (caudal) fascial structures to the most superior (cranial) tissues.

### 2.6. Risk of Bias in Individual Studies

The risk of bias (ROB) assessment was conducted independently by two reviewers (M.R. and M.P.C.) using the Anatomical Quality Assurance (AQUA) Tool [50]. This tool, endorsed by the International Federation of Associations of Anatomists, was designed to assess the quality of anatomical studies included in systematic reviews [41,50] using five key domains: (i) objective(s) and subject characteristics; (ii) study design; (iii) characterization of methods; (iv) descriptive anatomy; and (v) results reporting. Each domain utilizes a set of questions to characterize the ROB. These questions were scored as having a low, high, or unclear ROB and therefore, each related domain was classified as having a low, high, or unclear ROB. An overall decision on ROB was then made for each study according to the following criteria: (i) a study was considered as having an overall high ROB when high ROB was attributed to at least one domain or when some concern of bias was attributed to two or more domains; (ii) a study was thought to have some concern of bias when one domain was rated as having some concern of bias; and (iii) a study was deemed to be at low ROB when all domains were scored as low ROB [51].

### 2.7. Data Synthesis

The studies included were heterogeneous in nature owing to the varied methodology involved, thus rendering a meta-analysis and pooling of quantitative data unfeasible. Therefore, a critical synthesis of the information was conducted and quantitative data from individual studies were reported when applicable.

## 3. Results

### 3.1. Study Selection

The literature search yielded a total of 4354 studies from the aforementioned databases. After the removal of duplicates, 3710 articles were eligible for title and abstract screening. Of those, 108 articles were screened for full-text assessment and finally, 39 articles met the eligibility criteria (Figure 1). Overall agreement between reviewers for the final selection of articles included was 99.8%, thereby indicating excellent agreement [49].

### 3.2. Study Characteristics

The characteristics of the studies included in our review are presented in Table 1. The studies were published between 1971 and 2021. All studies included were observational studies, of which 87% (n = 34 studies) were cohort studies and 13% (n = 5 studies) were case-control studies. The assessment approaches used were as follows: 67% dissection (n = 26 studies), 36% histological techniques (n = 14 studies), 21% imaging (ultrasound or magnetic resonance (MRI)) (n = 8 studies) and 8% clinical morphological measurements (n = 3 studies). The studies included 1192 women, among which 57% (n = 679) were living females, 25% (n = 300) were embalmed cadavers and 14% (n = 165) were fresh cadavers. The remaining 4% (n = 48) were cadavers whose preservation technique was not specified. As illustrated in Table 2, the studied structures included the perineal membrane (PM) (n = 7 studies), the perineal body (PB) (n = 10 studies), the endopelvic fascia (n = 5 studies), the pubourethral and pubovesical ligaments (n = 10 studies), the tendinous arch of pelvic fascia (TAPF) (n = 10 studies), the tendinous arch of levator ani (TALA) (n = 4 studies), the pubocervical fascia (n = 3 studies), the rectovaginal fascia (RVF) (n = 9 studies), the tendinous arch of rectovaginal fascia (n = 1 study), the rectovaginal ligament (n = 1 study), the rectosacral and inferior fascia of the pelvic diaphragm (n = 2 studies), and the paracolpium (n = 2 studies). No studies reported the anatomy or composition of the membranous layer of the perineal subcutaneous tissue, perineal fascia, or anococcygeal ligament.

### 3.3. Risk of Bias (ROB)

The overall ROB was scored as low in 41% (n = 16), unclear in 8% (n = 3) and high in 51% (n = 20) of the studies. The ROB assessment for each study is presented in Table 3. For domain 1 (i.e., objectives and subject characteristics) 51% (n = 20) of the studies were assessed as having a low ROB, 44% (n = 17) of the studies had an unclear ROB and 5% (n = 2) of the studies were scored as having a high ROB. As for domain 2 evaluating the study design, 76% (n = 28) were considered as having a low ROB, 21% (n = 8) an unclear ROB and 3% (n = 1), a high ROB. Domain 3 examined the characterization of methods with 69% of the studies (n = 27) being assessed as having a low ROB, 21% (n = 8) an unclear ROB and 10% (n = 4) a high ROB. In terms of domain 4, which relates to descriptive anatomy, 62% (n = 24) of the studies presented a low ROB, 31% (n = 12) an unclear ROB and 7% (n = 3) a high ROB. Lastly, for domain 5 pertaining to results reporting, 56% of the studies (n = 22) were scored as having a low ROB, 26% (n = 10) an unclear ROB and 18% (n = 7) a high ROB.

### 3.4. Anatomical Structures Described

#### 3.4.1. Perineal Membrane (PM)

Seven studies [54,55,63,65,66,67,70] involving a total of 145 females (24 living and 121 cadavers) reported the anatomy and organization of the PM. The range of ages reported in the studies was 22 to 104 years of age. Three studies [58,67,68] used a dissection approach, three [66,67,72] used a histological approach (alone or combined with dissection), and two [54,67] employed MRI to study the anatomical course of the PM. Almost all of the studies were consistent in describing the PM as a complex structure embedded into a larger interconnected system [55,63,65,66]. The structure was further described as dense triangle-shaped connective tissue extending between the inferior borders of the ischiopubic rami (the lateral margins of the triangle). The anterior margin of the PM was found to merge with the TAPF at the level of its insertion on the pubic bone. The posterior margin is connected medially to the perineal body [63,65,66]. The PM was shown to be perforated by the urethra and the vagina in its center. Using both dissection and MRI approaches, this structure was described as being divided into anterior and posterior portions [65,66,67] presenting with distinct morphologies and functions. The anterior portion was observed as a cubic mass containing several interconnected structures. From a transverse plane perspective, the anterior portion of the PM was continuous in its midline with the paraurethral and paravaginal connective tissue [65,66,67]. The compressor urethrae and urethrovaginal sphincter muscles were both embedded and interdigitated with the PM. A connection with the bulbospongiosus muscle was also observed [65,66,67]. From a sagittal plane perspective, the superior surface of the anterior portion of the PM connected with the inferior fascia of the levator ani muscles (LAM), whereas the inferior surface fused with the crura of the clitoris and the bulbs of the vestibule [65,66,67]. The morphology of the posterior part was described as a band of fibrous connective tissue that connected to the lateral walls of the vagina and to the ischiopubic rami. The range of thicknesses measured for the PM was between 3 mm and 12 mm [65]. Brandon, et al. [67] confirmed that the PM and its relationships with the vestibular bulb, clitoral crus, and LAM could be clearly seen with an MRI in living females.

From a histological point of view, the composition of the PM varied depending on location (i.e., anterior vs. posterior portions). A predominance of elastic fibers was observed by Kato, et al. [65], who also found collagen fibers related to the epimysium of the compressor urethrae and urethrovaginal sphincter muscles. In the anterior portion, the fibers were horizontally oriented. However, this alignment changed to a more oblique and inferior orientation in the posterior portion, becoming almost vertically oriented along the longitudinal axis of the vagina, where the fibers appeared more densely packed [63,70]. Embedded striated muscle fibers belonging to the compressor urethrae and urethrovaginal sphincter muscles were observed in the anterior portion [63,65]. In the posterior portion, Betschart, et al. [63] consistently found smooth muscles. Striated muscles were observed inconsistently and were less abundant. Smooth muscle fibers were also found in the posterior portion in other studies [65,66]. Striated muscle fibers inserting into the posterior portion were identified as belonging to the LAM [65,80]. Other structures such as blood vessels and adipose tissue were identified in the vicinity of the PM [63,65,66].

#### 3.4.2. Perineal Body (PB)

Ten studies [55,62,65,68,71,73,74,75,76] assessed the anatomy of the PB in a total of 604 females (485 living and 119 cadavers) aged between 15 and 104 years of age. Two studies [55,75] used a dissection approach, one [76] used a histological approach only, two [62,65] combined dissection and histology, one [68] employed MRI, one [73] performed ultrasound imaging, and three [73,76,81] utilized clinical morphometrical measurements to describe the PB. All studies were consistent in characterizing the PB as a mass of fibromuscular tissue lying between the anal canal and the vaginal wall. Several authors described the pyramidal shape of the PB, which had a large base located inferiorly underneath the perineal skin and a tip extending superiorly between the vagina and the anorectum [70,75,77]. The PB was also shown to connect the two halves of the membrane together [55]. The PB was also observed to be in continuity with the RVF [62]. The PB was commonly described as having a superior and inferior portion [58,65,75,77]. Some authors also characterized the lateral expansions of the PB more specifically in relation to muscle insertions [58,65,77]. All dissection and histological studies reported that the PB was the site of insertion of several pelvic floor muscles [58,65,67,77,78]. With the use of MRI, Larson, et al. [68] were able to visualize the PB and precisely identify the individual muscle insertions. In contrast with the other studies delineating a superior and inferior portion, this imaging study described three portions: an inferior portion, a midportion, and a superior portion. The inferior portion comprised the attachments of the bulbospongiosus muscle, the superficial transverse perineal muscle, and external anal sphincter. The midportion was composed of the insertions of the superior aspect of the transverse superficial muscles, the pubovisceralis muscle (e.g., puboperinealis and puboanalis) as well as the internal and external anal sphincter muscles. The superior portion encompassed the internal anal sphincter as well as some portion of the pubovisceralis (e.g., pubovaginalis and puboanalis). The puborectal muscle was not found to contribute to the PB as it formed a loop behind the rectum. Various PB lengths (i.e., anteror-posterior distance), measured from the posterior fourchette to the mid-anus, were reported in many studies [65,73,76,81]. The mean PB length assessed in young primigravid Cameroonian women at 36 weeks of pregnancy was 3.21 cm (SD 0.75; range 1.5–5.5). Measured during the first stage of labor in primigravid women, the mean PB length obtained was 3.7 cm (SD 0.5; range 2.3–5.0) [74] and 3.9 cm (95%CI: 3.8–4.0; range 2.2–6.0 cm) [71] in samples mainly from Caucasian women and multiracial women, respectively. In a study using a dissection approach in older embalmed cadavers, a smaller length was measured, ranging from 1 cm to 2 cm [62]. The measure also appears to be smaller (mean length of 1.45 cm (SD 0.15)) in the study by Santoro, et al. [73] in nulliparous middle-aged women using 3D endovaginal ultrasound. This imaging approach not only enabled the length of the PB to be assessed but also the width (mean width: 1.33 cm (SD 0.12)) and thickness (mean thickness: 0.76 (SD 0.05)) [73]. All histological studies [65,67,78] detected the presence of smooth muscle fibers and blood vessels in the PB. Kochová, et al. [76] and Kato, et al. [65] found that the PB also contained skeletal muscle fibers, collagen fibers, elastin fibers, adipose tissue, and peripheral nerves.

#### 3.4.3. Endopelvic Fascia

Five studies [16,36,57,58,71] investigated the anatomy of the endopelvic fascia with a total sample size of 136 females (4 living and 132 cadavers). Three studies [16,36,58] performed dissections on cadavers to explore their fascial anatomy. A different study [69] used a histological approach, and another study [54] used MRI. Although the terms “endopelvic fascia” and “parietal fascia” were sometimes used interchangeably, most studies agreed that it was one continuous unit with a visceral component and a parietal component. The visceral component referred to the fascial lining that surrounded the pelvic viscera. The parietal component is related to the internal lining of the pelvic floor and the walls of the pelvis. More specifically, the authors observed that the parietal component covered the LAM, internal obturator, ischiococcygeus, and piriformis muscles [34,70]. The portion overlaying the LAM was named the superior fascia of the pelvic diaphragm. The parietal component also covered the anterior surfaces of the sacrum, coccyx, and pelvic walls to delineate the pelvic cavity [16,36,57,58]. Moreover, parts of the endopelvic fascia that attached to the urinary bladder, uterus, vagina, rectum, and to the pelvic walls were also recognized as specific areas of endopelvic fascia [16,36,57,58]. Hirata, et al. [69] observed an abundance of elastic fibers running perpendicularly to the LAM, with little to no smooth muscle fibers, except for the area close to the rectovaginal fascia.

#### 3.4.4. Pubourethral and Pubovesical Ligament

Ten studies [17,18,19,22,24,34,52,54,59,61] examined the anatomy of the pubourethral ligaments, combining a total of 181 females (41 living and 109 cadavers). The age range across studies was 19–106 years. Eight studies [18,19,22,24,36,55,62,64] conducted dissections, five [18,19,22,24,64] used a histological approach, and three [17,18,57] used MRI. The pubourethral ligaments were described as a band of connective tissue that originated from the posterior surface of the pubic bones in the area of the pubic symphysis [18,19,24,36,55,57,62] and inserted on the urethra and anterior vaginal wall [18,24,57,62]. Two studies [17,18] reported MRI data of the pubourethral ligament on cadavers and living females and one [61] pubovesical muscle of living females. Four studies [19,22,36,62] described the pubourethral ligament as being one structure divided into two distinct portions (i.e., superior-inferior). Three studies [17,24,55] mentioned that it was divided into three portions (i.e., superior-intermediate-inferior), while others found that this structure comprises two different ligaments (i.e., pubourethral and pubovesical ligaments) [17,19,36]. El-Sayed, et al. [18] were able to visualize the superior pubourethral ligament in living females on MRI [61]. In contrast, one study [61] refuted the existence of ligaments in this area and instead proposed that the structure running between the pubic symphysis and urethra are composed of muscle fibers corresponding to the pubovesical muscle. Mean lengths were reported for the superior pubourethral ligament [24] (10.0 mm) and for the pubovesical ligament (12.3 mm) [17]. The composition was shown to vary depending on the area: the inferior part of the pubourethral ligament was shown to be rich in smooth muscle fibers [17,19,24], the intermediate portion was composed of collagen with few capillaries and scattered muscle fibers [18,24], and the superior part was rich in dense collagen fibers.

#### 3.4.5. Tendinous Arch of Pelvic Fascia (TAPF)

Ten studies [17,23,34,53,56,58,59,60,67,70] focused on the anatomy of the TAPF and involved a total of 166 females (30 living and 136 cadavers). The age range across studies was 14–94 years. Seven studies [23,36,56,57,59,61,63] used a dissection approach, one study [70] used a histological approach, and two studies [17,67] employed MRI to explore the anatomy of the TAPF. In most studies, the TAPF was described as a thickening of the endopelvic fascia. The TAPF looked like a band of connective tissue that originated from the infero-medial part of the posterior surface of the pubic symphysis and inserted onto the ischial spine [17,34,53,56,58,59,60,67]. Anteriorly, the TAPF was continuous with the anterior portion of the PM and covered the medial part of the LAM, and it fused with the paraurethral and paravaginal connective tissue. On its most posterior part, it connected with the tendinous arch of levator ani (TALA) and ran onto the internal obturator muscle. A single study [23], conducted in only 5 cadavers, described the TAPF as a discrete and inconsistent structure. In terms of its morphometry, several studies [17,23,63] observed a mean length of 8.1–9.0 cm, which was deemed shorter than the common clinical measure of 10.0 cm reported by gynecologists [60]. Furthermore, DeLancey [53] highlighted that it inserted 1 cm above and lateral to the lower border of the pubic symphysis (without specifying if that measurement referred to a mean), while Pit, et al. [59] reported a site located at an average of 4 mm (range 3–8 mm) lateral to the pubic symphysis. The TAPF could be clearly visualized with MRI [67]. A single study [70] investigated the TAPF using a histological approach. That study observed that the TAPF was a mesh of connective tissue largely composed of collagen fibers with an abundant presence of elastic fibers running along the mediolateral axis in the transverse plane. Smooth muscle fibers tended to be more densely packed posteriorly near the RVF but scattered anteriorly near the pubocervical fascia. The composition of the TAPF was found to be similar in nulliparous and multiparous women [70].

#### 3.4.6. Tendinous Arch of Levator Ani (TALA)

Four studies [17,23,36,62], which included a total of 59 females (10 living and 49 cadavers), investigated the anatomy of the TALA. Of these, three studies [23,36,62] used a dissection approach and one study [17] used MRI. Studies included in this review [17,23,36,62] reported that the TALA was a fibrous band that originated anteriorly on the superior aspect of the pubic rami bilaterally [23,59]. Most studies confirmed that the TALA inserted onto the ischial spine area [17,36,62] whereas only one study [23] found the insertion on the sacrum, but did not discuss this discrepancy in comparison with the other authors. In addition to the ischial spine insertion, one study [34] added that the TALA merged with the mid-portion of the TAPF to join the ischial spine. Although not systematically assessed, some studies [23,36,62] also reported on the intimate relationship between the TALA and the LAM. Ersoy, et al. [23] showed that the TALA attached to the LAM and could be considered a thickening of the internal obturator fascia. Further describing the connection with the LAM, Pit, et al. [59] noticed that the TALA formed the superior border of the connective tissue plate from which the medial side of the posterior part of the pubococcygeus muscle and iliococcygeus muscles originated. Using MRI, Li, et al. [17] were the only ones to report morphological details of the TALA with a mean length of 4.8 (SD 1.7) cm. No studies described the histological composition of the TALA.

#### 3.4.7. Pubocervical Fascia

Three studies [16,36,74] assessed the anatomy of the pubocervical fascia in a total of 111 female cadavers ranging in age from 26 to 104 years. Although all studies used dissection, Hinata, et al. [72] also used a histological approach. The pubocervical fascia was described as a structure lateral to the urethra and the urinary bladder and covering the anterior aspect of the vagina. It was found to originate from the pubic symphysis and to be suspended from the TAPF, to insert onto the lateral third of the vagina in order to provide support [16,34]. Histologically, a network of thick collagen fiber bundles could be identified within the lateral part of the pubocervical fascia; however, this structure did not contain smooth muscles [72].

#### 3.4.8. Rectovaginal Fascia (RVF)

Eight studies [12,16,20,21,36,60,74,79] involving a total of 250 females (102 living and 148 cadavers) investigated the anatomy of the RVF. The age range across studies was 15–104 years. Nine studies [12,16,20,21,36,60,74,79,80] used a dissection approach, four studies [12,20,21,74] used a histological approach, and one study [78] employed MRI. The RVF was described as fibrous tissue located between the rectum and the vagina. It was described as running in a coronal plane [12] attaching to the peritoneum of the recto-uterine pouch and to the PB. Some authors observed that it fuses to the fascia of the LAM [12,20,21,77,80]. Two studies [20,78] reported the mean length of the RVF. Using MRI, Rodriguez, et al. [78] measured a mean length of 73.2 mm (SD 15.3) from the posterior fornix and the end of the recto-uterine pouch (also called the pouch of Douglas) to the PB. Nagata, et al. [20] obtained measurements ranging from 35 mm to 60 mm from the recto-uterine pouch to the superior end of the external anal sphincter. Three studies investigated the thickness of the RVF [12,20,80]. Rodriguez, et al. [78] (MRI data) measured the thickness at the superior third (mean 2.8 ± 1.7 mm), at the middle third (2.2 ± 1.2 mm), and at the inferior third (2.5 ± 1.3 mm). Nagata, et al. [20] reported thickness measurements ranging from 0.1 mm to 0.3 mm, depending on the level of the vagina, using dissection. Lastly, Stecco, et al. [12] provided a more detailed morphometrical description by taking thickness measurements on 20 female unembalmed cadavers. Measures were taken in the midline and at the lateral aspect of the RVF, and at the level of the middle and lower third of the vagina. For both levels, their thickness values for the lateral part of the RVF were slightly greater than those of Rodriguez, et al. [78] (middle third: 2.67 ± 1.08 mm; inferior third: 2.64 ± 0.12 mm). Stecco, et al. [12] also observed that the RVF was thicker in its lateral aspect as opposed to its midline (middle third: 1.75 ± 0.75 mm; inferior third: 1.70 ± 0.88). The histologic composition of the RVF was described as a network of collagen, elastic, and smooth muscle fibers embedded in loose connective adipose tissue [12,20,21,74].

#### 3.4.9. Tendinous Arch of Rectovaginal Fascia

One dissection study [34] assessed the tendinous arch of the rectovaginal fascia and included a total of 30 fresh female cadavers with an age range of 48–92 years. The tendinous arch of the rectovaginal fascia was described as a taut band that ran between the PB and the middle third of the vagina and was also observed to merge with the TAPF. It also appeared as a line formed from the fusion of the RVF and the superior fascia of the pelvic diaphragm that covered the LAM.

#### 3.4.10. Rectovaginal Ligament

Like the tendinous arch of the rectovaginal fascia, the rectovaginal ligament was observed by Ercoli, et al. in the same study. The ligament originated from the posterolateral pelvic walls at the levels between the sacral foramina S2 to S4 and the ischial spines. The rectovaginal ligament was also described as a convergence of fibers of the endopelvic fascia and the visceral fascia that narrowed towards the inferior two-thirds of the vagina.

#### 3.4.11. Rectosacral Fascia and the Inferior Fascia of the Pelvic Diaphragm 

Of the two dissection studies [34,64] carried out in a total of 35 female cadavers, both evaluated the macro-anatomy of the rectosacral fascia and one [64], the inferior fascia of the pelvic diaphragm (also called the Waldeyer’s fascia) as a distinct structure from the rectosacral fascia. They reported that the rectosacral fascia originated from the presacral fascia at the level of S2, S3, or S4 and joined the fascia propria of the rectum 3–5 cm above the anorectal junction. This fascia was found to clearly divide the retrorectal space into inferior and superior portions [64]. As for the inferior fascia of the pelvic diaphragm, it was also described to originate from the presacral fascia, but in contrast, it was found to form the floor of the inferior portion of the retrorectal space [64].

#### 3.4.12. Paracolpium

Two studies [16,72] evaluated the anatomy of the paracolpium in a total of 71 female cadavers with an age range of 26 to 104 years. Although all studies [16,72] used dissection, Hinata, et al. [72] also used a histological approach. The paracolpium was characterized as a loose connective tissue that attached the vagina to the pelvic walls [16]. It was also shown to be continuous with the cardinal and uterosacral ligaments. One study [72] found that the paracolpium was rich in elastic fibers, with venous and nervous plexuses running into it.

## 4. Discussion

To our knowledge, this review is the first to present the current state of knowledge on the macro- and micro-anatomy of the female pelvic floor fasciae. Some narrative reviews [81,82,83,84,85,86,87,88,89] in the past have focused on specific fasciae or connective tissue structures in a particular area of the pelvic floor. However, a global and systematic search approach was needed to assess the integrated nature of this fascial system based on studies presenting original data. This systematic search and review yielded a total of 39 studies that combined 1192 women (679 living and 513 cadavers). The quality of anatomical studies included in this work demonstrated a combined unclear and high ROB as high as 92%, resulting in only 8% of the reviewed articles being considered as having a low ROB. It is clear from this review that the complexity of the fascial anatomy reaches a paroxysm in the pelvic region. These fasciae are most often interconnected, organized in juxtaposed and nested layers that are oriented in different directions. This review sheds light on the anatomy of the PM, PB, endopelvic fascia, pubourethral and pubovesical ligaments, TAPF, TALA, pubocervical fascia, RVF, retrosacral fascia, and the inferior fascia of the pelvic diaphragm, and paracolpium in women. We also reported that the PM, PB, TAPF, TALA, pubourethral ligaments, pubovesical ligament, and RVF can be clearly visualized with an MRI assessment. Trying to understand the anatomy, organization, and composition of what should be considered the normal fascial anatomy of the female pelvic floor could be helpful in determining what could account for underlying pathological mechanisms and thereby developed, adapted, and targeted treatment modalities.

### 4.1. Membranous Layer of the Perineal Subcutaneous Tissue 

The membranous layer of the perineal subcutaneous tissue (also called the Colles’ fascia) is the most inferior structure (or the most superficial) described in traditional anatomy textbooks [39,46,47]. This structure is also called the membranous layer of the subcutaneous tissue of the perineum. Even though these related terms were included in our search strategy, no studies focusing on its macro- or micro-anatomy were retrieved. This suggests that even if the subcutaneous tissue has been described in textbooks [39,46,47], its anatomy, topographical organization, and histology are far from being fully understood. The presence of a distinct layer that is in continuity with a membranous layer of the abdomen called the Scarpa fascia has been described by anatomists and surgeons [90,91]. In other areas of the body, this tissue has been characterized as a loose areolar connective tissue composed of cuboid and flat adipose cell lobules and a mix of interwoven collagen and elastin fibers [42,92,93]. Among the different models proposed describing its architecture, the skin ligaments (retinacula cutis) running between the adipocyte-rich lobules of the subcutaneous tissue form a three-dimensional network, providing a dynamic anchor of the skin to the underlying tissue. The retinacula cutis is the endpoint of nerves, blood vessels, and lymphatic networks. All of these originate at deeper levels of the human body and travel out to the surface to provide nutrition and sensory innervation to the skin. However, while this architecture has been shown by some authors in non-pelvic areas [37,91], others rather suggest a fascial continuum model composed of inseparable connective tissue units with no discontinuity [94,95]. Since no data specific to the membranous layer of the perineal subcutaneous tissue could be retrieved in this review, we cannot confirm or refute a particular tissue organization and composition. This gap in knowledge helps to stimulate and nourish the debate on the existence or non-existence of a distinct membranous layer that can be conceptualized as “a true fascia”. Macro-and micro-anatomical studies combined with emerging imaging technologies focusing on this region would help to shed light on this topic. As for perineal fascia (also called the Gallaudet’s fascia), no specific study highlighting this structure was identified in our search. This fascia, or its related terms such as the deep perineal fascia or the superficial investing fascia of perineum [35,96], has been described by experts as covering the bulbospongiosus, the ischiocavernosus, and superficial transverse perineal muscles when present. It is attached laterally to the ischiopubic rami and fused anteriorly with the suspensory ligament of the clitoris. It is continuous anteriorly with the deep investing fascia of the abdominal wall. It is worth mentioning that this description is supported by very few studies reporting original data [92,93], and that this continuity is mainly derived from anatomy textbooks [97,98,99] or has been deduced from the literature [35,96].

### 4.2. Perineal Membrane (PM)

The PM, previously called the urogenital diaphragm by Luschka [100] in 1864, was described as a layer of striated muscle covered by a superior and inferior fascial layer extending between the ischial tuberosities. Far from the traditional simplistic description of a trilaminar fascia-muscle-fascia structure [66], several studies [65,66,67] in our review were consistent in describing the PM as either having interdigitations with certain muscular and connective tissue structures or all structures embedding others. This organization is in accordance with a more integrated and functional vision of the anatomy. This explains the heterogeneity of our findings given that different areas of the PM, mainly its anterior and posterior portions, presented different morphologies (and probably different thicknesses), as well as varied fiber type compositions and organizations. According to Stein [66], although the term “membrane” is now the commonly accepted nomenclature, seeing this structure as a sheet contributes to the misunderstanding about its anatomy and function. The horizontal orientation of the elastic fibers [65] and the connection between the PM and the epimysium of the compressor urethrae and urethrovaginal sphincter muscles lead us to believe that the anterior portion is mainly associated with urethral support. In the posterior portion, the connection with the PB and the paravaginal connective tissue, the vertical orientation of the fibers along the vaginal axis, and the direct insertion of the LAM indicates that this portion of the PM seems to be associated with the support of the vagina and the PB.

In regard to its precise histological composition, Betschart, et al. [63] did notice the predominance of connective tissue, but they were not specific about the fiber types. The predominance of elastic fibers was consensual, but the collagenous content description was not so clear. Hirata, et al. [70] and Betschart, et al. [63] showed that collagenous fibers run along the PM, but Kato, et al. [65] reported that the collagen fibers were limited to the perimysium of the LAM. These discrepancies might be attributed to the location of tissue samples. Betschart, et al. [63] observed a predominance of smooth muscles in the PM, while Hirata, et al. [69] found that smooth muscle was rare or absent. The presence of smooth muscle is in accordance with other studies [37,101] on the histology of fasciae in other areas of the body, which previously confirmed the presence of smooth muscle. The PM is therefore not different from any other fasciae in that sense. The fact that the presence of striated muscles was inconsistently reported may be explained by the age of the cadavers. Studies showed that all layers of striated urethral muscle become thinner in older women and may even disappear with aging as does nerve density [13,14,17,19,102]. Interestingly, the striated muscle fibers that were found [65,66] were not attributed to the deep transverse perineal muscle. This is in line with the review by Mirilas [88] who concluded that this muscle is not present in women. It is worth noting that Aronson, et al. [54] in 1995 failed to describe the PM using MRI. This could be explained by the use of a previous technology since Brandon, et al. [67] were able to correlate the same findings using MRI in 2009 that were reported in dissection studies.

With the evidence collected in this review and considering that the majority of the studies presented low ROB, we can confirm that the PM is neither a simple membrane nor a three laminar fascia-muscle-fascia unit. It is a complex multi-structure tissue integrated into a larger interconnected system, and thus, its histological composition varies according to the different structures. This highlights the importance of specifying the location of samples in future studies. There is clear evidence of the presence of distinctive anterior and posterior portions that might play different roles. The PM is mainly composed of dense connective tissue with a predominance of elastic fibers. The striated muscle embedded into the PM is not the deep transverse perineal muscle. The term “membrane” seems reductive to certain authors. Therefore, clearly defining the PM as a fascia could possibly better reflect the complexity of this structure. We identified some gaps in knowledge regarding the PM. Little is known about the somatosensory receptors present and the density and components of ground substances such as hyaluronan. This is relevant as it is now recognized that hyaluronan contributes to the viscous properties of fasciae [103,104] and therefore might play a role in the mechanics of the urogenital system. More recently, hormonal and cannabinoid receptors have been found in fasciae of other parts of the body [105] and it would be relevant to investigate if they are present in the PM. Filling these gaps would be helpful in deepening our understanding of conditions like vulvodynia. Moreover, little is known about the dynamic interaction between the different components of the PM. For example, trauma resulting from childbirth could have an impact on the integrity of the posterior portion of the PM.

### 4.3. Perineal Body (PB)

The PB is the structure that has drawn the greatest empirical attention with 10 studies included in the review involving a total of 604 women. The popularity of this structure is probably attributable to its accessibility, facilitating dissection, histologic and imaging assessment along with its proposed key role in the static pelvis [55,106]. It has indeed been suggested that the PB limits the opening of the urogenital hiatus and has an important anchor role for the vagina and the anorectum [55,106]. Studies included in the current review are consistent in describing the PB as a fibromuscular pyramid-shaped structure located between the vagina and the anorectum [65,70,75,77]. This node located in the middle of the perineum is the central crossroad for different layers of fasciae and muscle insertions. All layers of the pelvic floor muscles, namely the bulbospongiosus, the superficial transverse perineal muscle, the internal and external anal sphincter, and the LAM (i.e., the pubovisceralis muscle) were shown to converge to insert into the PB. These findings arose from dissection and histological studies [58,65,67,77,78] as well as from a low ROB MRI study [68] describing what was more precisely the muscle insertions. The latter study joined the aforementioned debate about the existence of the deep transverse perineal muscle because the authors did not visualize this structure attaching to the PB [68]. As for the fascia connections, the studies included showed that the PB was in continuity with the PM and the RVF [55,62]. The relationship between the PB and the membranous layer of the perineal subcutaneous tissue was not described in the studies retrieved, although a description had been proposed by experts [35]. Given that the PB was shown to have lateral anchors to the PM as well as superior extensions through the RVF, DeLancey, et al. [55] emphasized the supporting role of the PB. It was discussed that the PB provides support to level II and III of the pelvic suspensory system, which corresponds to the inferior two-thirds of the vaginal vault.

Notwithstanding the popularity and relevance of the PB, there are still some uncertainties regarding its morphometry. A wide range of PB lengths (1.0–6.0 cm) has been reported in the studies included in this review [65,73,76,81]. This variability may be attributable to the heterogeneity of the methods employed (i.e., measurements taken using MRI or ultrasound images that allow clinicians to delineate the structure of the PB vs. cadavers embalmed using different techniques vs. assessment in living women), including the anatomical landmarks used to measure the PB. DeLancey [55] also stressed the fact that there could be a high likelihood of errors when using cadavers due to embalming artifacts, loss of muscle tone, and distortion created by dissection, which can affect the reported anatomy of structures like the PB. The characteristics of the sample including age, BMI, ethnicity, and parity may also explain the discrepancies as these characteristics may affect tissue composition [58,65,67,78] and therefore, the PB’s length and thickness. The exact influence of these factors remains unclear as only two studies with high ROB have been conducted [62,71]. 

Regarding the histological content, the PB was found to be predominantly composed of collagen and adipose tissue, followed by elastin, smooth muscle, and peripheral nerves [65,76]. It is unclear if the content of the PB varies according to the different sites of the PB as Soga, et al. [62] found different tissue content in the lateral extensions of the PB while Kochová, et al. [76] obtained similar composition results across 5 sites of the PB in a volume faction analysis. Nevertheless, the tissue composition observed in the studies included in this review aligned with the recommendation of Terminologica Anatomica [47] to use the term PB instead of centrum tendineum perinei. The PB is indeed a fibromuscular structure rather than a centrum tendineum perinei as previously reported [107]. 

### 4.4. Endopelvic Fascia

There is an ongoing debate about which structures are considered part of the endopelvic fascia and whether those structures are distinct structures. In alignment with the studies of this review, we can support the paradigm stating that the endopelvic fascia is a continuous unit with various thickenings or condensations that have been named fasciae or ligaments. The endopelvic fascia lines the pelvic cavity, including its floor and its walls, the pelvic viscera, and their attachments to the pelvis [16,34]. The concept that ligaments, namely the broad, uterosacral, and cardinal ligaments, might be a reinforcement of certain parts of a fascia that are subject to tensional forces (e.g., the weight of the pelvic organs), has been proposed for other parts of the body [37]. Stecco, et al. [37] described the inguinal ligament as being a thickening of the distal part of the fascia of the external oblique muscle as it joins the fascia lata. Ercoli, et al. [34] described the endopelvic fascia to encompass the internal obturator and the piriformis fascia. This is in agreement with Ramin, et al. [35], who proposed a continuity model of the endopelvic fascia not only with the abdominal wall fasciae, but also with those of the lower back and hip, including the fascia of the internal obturator. This model combined with the evidence retrieved in this review highlight the possible mechanical link between visceral and musculoskeletal kinematics.

Knowledge of the histological content of the endopelvic fascia is scarce. In a low ROB study, Hirata, et al. [69] showed that the female endopelvic fascia is a monolayer connective tissue structure rich in elastic fibers, running in an anteroposterior direction and across the LAM fibers. They detected very few or no smooth muscle fibers, with the exception of the most posterior area near the rectum, which exhibited a higher density of smooth muscle fibers. This morphology and composition were similar between nulliparous and multiparous women [70]. It is plausible that the composition of the tissue may differ in parts of the endopelvic fascia due to organ support. For example, the TAPF, which could be conceptualized as a particular region of the endopelvic fascia, is predominantly composed of collagen fibers. In addition, the predominance of elastic fibers is very relevant from a functional standpoint. The pelvic organs move in two ways in order to function in an optimal manner: (1) by their own intrinsic active movement (e.g., motility) and (2) by the pull and push of the neighboring tissues. For mobility to occur freely, the endopelvic fascia must exhibit elastic strain and this property is provided by the elastic fibers. 

In summary, the endopelvic fascia is one continuous structure and its morphology and composition vary according to the anatomical region. Histological analysis has been limited to fiber-type content. Because of its role in organ support, it would be interesting to know more about sensory innervation: Would the so-called ligaments provide proprioceptive information like the ligaments of the musculoskeletal system do? Knowing more about the nociceptor content would provide insight into the contribution of this fascia to pelvic pain. The mechanical properties and the role of endopelvic fascia in a broader system involving the fasciae of the abdomen and lower limb need to be investigated with robust empirical data.

### 4.5. Particular Areas of the Endopelvic Fascia

The structures presented below are described in current textbooks as distinct and independent tissues. However, our review highlighted the difficulties encountered in clearly delineating their margins/boundaries as these structures are intimately related to (or are even in continuity with) the endopelvic fascia. We, therefore, chose to regroup them under the particular areas of the endopelvic fascia.

#### 4.5.1. Pubourethral and Pubovesical Ligaments

The pubourethral and pubovesical ligaments have been an area of debate for decades. In a recent review of the literature, Jepperson, et al. [32], in 2018, highlighted that even if the pubourethral ligament “term” is commonly used, there is no microscopic or histologic evidence that it is a different structure than the widely accepted Terminologica Anatomica term TAPF. These debates could partly reside in the fact that most of the studies (86%) related to these ligaments presented a high ROB. Milley and Nichols [52] were the first to refute the use of the term pubovesical ligament based on the assumption that this structure is not connected to the urinary bladder. Later on, DeLancey [19] described two distinct structures running from the area around the neck of the urinary bladder to the pelvic walls, stating that the pubovesical ligament was different from the pubourethral ligament. This description was convergent with the findings of Ercoli, et al. [34] and Li, et al. [17], the latter asserting that the pubovesical ligament was an extension of the detrusor muscle and played a role in miction by assisting in vesical neck opening. However, El Sayed [18] and Vazzoler [24] refuted the term pubovesical ligament and instead described three pairs (on either side of the midline) of pubourethral ligaments having three different portions (superior, intermediate, and inferior). Fritsch, et al. [61] were the only ones that denied the existence of a so-called “pubovesical ligament” and stated that it should be called the pubovesical muscle because its composition is mainly made of smooth muscle cells.

In terms of histology, the presence of collagen, elastin, and smooth muscle fibers was detected [17,18,19,24] with varying density depending on the specific portion of the ligaments. The variation in density was probably associated with a particular function, i.e., urethral support vs. urinary bladder opening during miction. Hamner, et al. [22] recently revealed the presence of fibroadipose tissue, blood vessels, nerves, and the fact that no structure was consistent with a discrete fascial layer. Our opinion, forged from the studies of this review, is that like any other ligament related to the support of an organ, the pubourethral ligaments can be seen as a thickening of the endopelvic fascia; they originate from the TAPF [19,36,55,57,62] and then follow different routes (urethra or urinary bladder neck). Evidence included in this study confirms that the superior [17,18] and inferior [18] portions of the pubourethral ligaments can be clearly visualized on an MRI scan [17,18] and that hopefully, these imaging technologies will help shed light on this structure in the near future. Histologic studies focusing on the sensory innervation content of these ligaments would provide more clarity on their function.

#### 4.5.2. Tendinous Arch of Pelvic Fascia (TAPF)

The TAPF, a thickening or a specialized part of the endopelvic fascia [17,34,53,56,58,59,60,67], was one of the most frequently described anatomical structures of the female pelvis in the studies included in this review. This is not surprising as this structure is related to various surgical procedures for incontinence and prolapse. Having a thorough understanding of its precise anatomical course is clinically relevant [60]. The findings of most studies agreed with its course, morphology, and length. The descriptions provided by the studies included in this review confirmed that the TAPF is a distinct structure from the TALA [53,59] and that the term “white line” should be reserved for the TAPF [14,56,62].

#### 4.5.3. Tendinous Arch of Levator Ani (TALA)

Some controversies still persist about the precise origin and insertion of the TALA, and whether or not it is a thickening of the internal obturator fascia or of the endopelvic fascia. Ersoy, et al. [23] observed in some specimens that the TALA was a thickening of the internal obturator fascia, whereas in other specimens it was an independent, stand-alone structure. The anatomical variations that the authors encountered could constitute an explanation for some of the controversies regarding this structure. However, unfortunately, since their sample size was very small (n = 5), further well-designed studies with low ROB are needed to resolve the discrepancy in anatomy as well as to investigate the histological composition of this structure.

#### 4.5.4. Internal Obturator Fascia

No single study included in this review specifically reported the anatomy and composition of the internal obturator fascia. However, some studies [23,34] observed that the internal obturator fascia was a thickening of the endopelvic fascia. Pit, et al. [59] clearly expressed their disagreement regarding that concept, arguing that the endopelvic fascia was only covering the medial part of the internal obturator muscle. Stecco and Hammer [37] described a distinct fascia enveloping the internal obturator and gemelli muscles and originating in the pelvis as a continuation of the iliac fascia. On the whole, these data suggest that the lateral part of the internal obturator is covered by an epimysial fascia that merges with the endopelvic fascia at the TAPF.

#### 4.5.5. Pubocervical Fascia

The pubocervical fascia is an important support structure for the urinary bladder, the urethra, and the vagina. It is often discussed in descriptions of surgical procedures, but only three studies investigating its anatomy were retrieved [16,36,74]. These studies described this fascia as extending from the lower part of the pubic symphysis and running along the anterior and lateral (middle third) vaginal wall. Laterally, it is connected with the pubococcygeus and suspended to the TAFP. According to DeLancey [16], the pubocervical fascia (anteriorly) and the rectovaginal fascia (posteriorly) attach the medial third of the vagina to the pelvic walls, to give an “H-shape” to the vagina at that level. However, although fibroadipose tissue was found between the anterior aspect of the vagina and the posterior aspect of the urinary bladder, none of these studies could solve the controversy over the existence of a separate membranous layer of fascia between the urinary bladder and vagina.

#### 4.5.6. Rectovaginal Fascia (RVF)

The existence of the RVF in women and the understanding of its anatomy has been a topic of debate for many years among anatomists and surgeons [12,83]. The main point of divergence is whether a distinct membranous fascial structure, analogous to the Denonvillier’s fascia found in men, exists between the vagina and the rectum in women, and whether the rectovaginal space is merely filled with a mix of adipose tissue, fragmented membranous connective tissue, and muscle fibers. This discordance may come from varying theories about its embryological origin that have been advanced: (1) the peritoneal fusion of the embryonic cul-de-sac theory, (2) the condensation of embryonic mesenchyme theory, and the mechanical pressure theory (for a review, see [12,83]).

When considered as a whole [77], the findings of the low ROB studies included in this review converge towards the existence of a distinct RVF layer located close to the posterior wall of the vagina. Although collagen and muscle fibers connected to the posterior wall of the vagina were observed, they did not connect with the rectum [57,59]. The RVF was indeed found to be separate from the anterior rectal wall by a layer of adipose connective tissue that surrounds the rectum and allows the two organs to move independently. Inferiorly, the RVF originates from connective tissue fibers of the PB. Then it runs cranially and laterally to attach to the pelvic sidewall via its fusion with the TAPF, approximately midway between the pubic symphysis and the ischial spine [77]. Superiorly, it is connected to the peritoneum. Evidence provided in this review confirms that the RVF can be clearly visualized and measured with MRI [78] and the findings correlate with the dissection observations. The RVF has been qualitatively described as a relatively strong membrane that can play a role in the transmission of force between perineal structures and organs, as does other fasciae in the body [102]. Therefore, trauma to this structure may impact the support of the organs and function of the pelvic floor muscles. The very complex mechanical interplay between these structures requires further investigation. Moreover, we still need to clarify if the RVF presents morphological and structural variations associated with age, hormonal status, or measurement sites that would explain the variability in thicknesses observed among the studies.

#### 4.5.7. Rectosacral Fascia and Inferior Fascia of the Pelvic Diaphragm

Controversies still abound about the concept of inferior fascia of the pelvic diaphragm (also called the Waldeyer’s fascia) and rectosacral fascia at the posterior limit of the retrorectal space. The inferior fascia of the pelvic diaphragm and rectosacral fascia are often thought to be synonymous. The anterior limit of the retrorectal space was found to correspond to the posterior rectal fascia and its posterior limit, to the parietal presacral fascia. According to Garcia-Armengol, et al. [64], the rectosacral fascia is a distinct structure that generally arises at the level of S4 to separate the retrorectal space into a superior and an inferior portion. The inferior fascia of the pelvic diaphragm is, in fact, a structure that covers the inferior limit of the inferior portion of the retrorectal space, and is formed by the fusion of the rectal visceral fascia with the presacral parietal fascia, in their most inferior part. The variability of the origin of the rectosacral fascia can explain the confusion about its existence. Origins as high as S2 have been reported [64,108]. Consequently, if a dissection with a perineal approach does not proceed cranially enough, the rectosacral fascia can be missed. A clear understanding of the topographical organization of the retrosacral space is clinically relevant when considering surgeries such as abdomino-perineal resection of the rectum or postnatal repair for anal incontinence [109]. In conclusion, since only high ROB studies were available for the rectosacral fascia, well-designed studies are needed to shed light on the anatomical uncertainties remaining for that structure.

#### 4.5.8. Paracolpium

Surprisingly, little is known about the anatomy of the paracolpium. A better understanding of this structure may have an impact on routine pelvic organ surgeries, like radical hysterectomy, in which resection of the paracolpium is an essential part of the procedure. The paracolpium and parametrium are the connective tissues surrounding the vagina and the uterus, respectively [16,72]. The ureter can be used as a landmark separating the lateral and ventral paracolpium (beneath the ureter) and the ventral and lateral parametrium (above the ureter). In the midvagina, the paracolpium fuses with the pelvic fascia laterally to attached to the pelvic wall. Medially, the paracolpium is connected to the pubocervical fascia. A more comprehensive knowledge of the anatomy of the paracolpium can lead the path to new surgical techniques that can avoid the resection of this important supportive structure [110].

### 4.6. Limitations

Some limitations in our systematic review and search should be acknowledged. First, more than half of the studies included in our review were assessed as having a high or unclear ROB. This could be explained by the fact that some studies were conducted several years ago, prior to the formulation of international recommendations for improving the quality of anatomical studies. Our review hence highlighted the importance of conducting future studies while relying on current guidelines for reporting observational studies like the STROBE statement [111] and using commonly accepted terminology [32,33,34]. Second, our review specifically targeted connective tissue located inferiorly to the endopelvic fascia for relevance and feasibility purposes, which may limit the generalizability of our findings. Further reviews are needed to expand our knowledge on the structures located superiorly as well as to better understand the relationship or continuity between the pelvic floor fasciae and the connective tissue of the surrounding areas (i.e., abdomen, back, lower limbs). Third, our review revealed considerable variation in the macro- and micro-anatomy of the pelvic floor fasciae. The paucity and quality of the data retrieved are insufficient to fully capture the factors responsible for this heterogeneity. Several studies noted interindividual variations in the anatomy (e.g., age and parous status) [17,20,23,57,62,66,67,71,72,80]. Age, parity, ethnicity, the presence of pelvic floor disorders, and the timing of data collection (e.g., tissue harvesting near time of death) were suggested as potential factors influencing the anatomy [22]. However, the authors of this review recognize that the majority of the females (i.e., cadavers) used in this type of research are older in age, who may have had children, suffered from endometriosis, or experienced other age-related issues in life. The authors also acknowledge that the only way to have normal reference values would be to perform dissections on young nulliparous women, which is obviously a significant challenge from a clinical, and not to mention, ethical standpoint. 

## 5. Conclusions

This review provides a comprehensive portrait of the evidence pertaining to the pelvic floor fasciae. The studies included allow for an integrated description of the macro- and micro-anatomy of the pelvic floor fascia and more specifically the PM, PB, endopelvic fascia, pubourethral and pubovesical ligaments, TAPF, TALA, pubocervical fascia, RVF, retrosacral fascia, and the inferior fascia of the pelvic diaphragm. Our findings also unearth some areas of uncertainty to encourage future studies. This review will hopefully provide a strong basis for investigating the function of each structure in relation to its origin/insertion and tissue composition as well as interindividual variations. This will set the stage to better understand the pathophysiology of various pelvic floor disorders and hence, develop adapted and targeted treatment modalities.

## Figures and Tables

**Figure 1 life-11-00900-f001:**
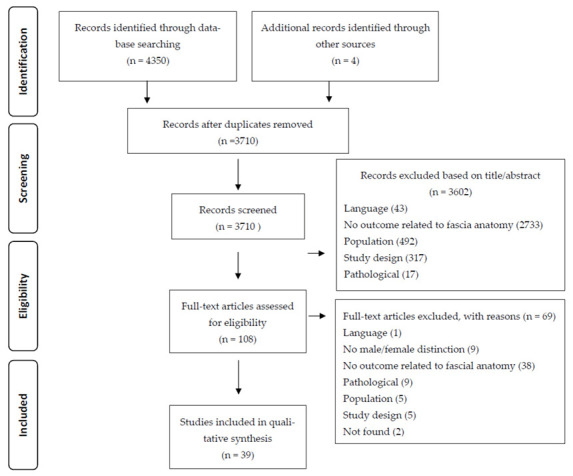
PRISMA Flowchart Diagram of the Selected Studies.

**Table 1 life-11-00900-t001:** Characteristics of studies evaluating female pelvic floor fascia anatomy.

Authors	Study Design	Assessment Approach	Sample	Age, Years Mean (SD or Range)
Milley and Nichols [52]	Observational study (cohort)	Dissection	14 embalmed female cadavers	N/A (range 22–91)
DeLancey [19]	Observational study (cohort)	Dissection + Histological	14 embalmed and 14 fresh female cadavers	53 (range 28–78) 50–60 (range 46–106)
DeLancey and Starr [53]	Observational study (cohort)	Dissection	2 embalmed and 2 fresh female cadavers	N/A (range 27–74)
DeLancey [16]	Observational study (cohort)	Dissection	19 embalmed and 42 fresh female cadavers	N/A (range 0–104)
Aronson, et al. [54]	Observational study (case-control)	Radiological (MRI)	4 living females (continent) (and 4 incontinent living females)	33 (range 29–38)
DeLancey [55]	Observational study (cohort)	Dissection	22 embalmed and 42 fresh female cadavers	N/A (range 0–104)
Mauroy, et al. [56]	Observational study (cohort)	Dissection	25 embalmed female cadavers	N/A
Leffler, et al. [57]	Observational study (cohort)	Dissection	10 embalmed and 2 fresh female cadavers	N/A
Occelli, et al. [58]	Observational study (cohort)	Dissection	2 fresh female cadavers	66 and 88
Vazzoler, et al. [24]	Observational study (cohort)	Dissection + Histological	8 embalmed and 2 fresh female cadavers	N/A (range 60–102)
Pit, et al. [59]	Observational study (cohort)	Dissection	10 embalmed female cadavers	N/A
Ersoy, et al. [23]	Observational study (cohort)	Dissection	5 embalmed female cadavers	61 (range 52–71)
Albright, et al. [60]	Observational study (cohort)	Dissection (including measures)	23 embalmed and 7 fresh female cadavers	N/A (range 64–94)
Ercoli, et al. [34]	Observational study (cohort)	Dissection	30 fresh female cadavers	67 (SD 8) (range 48–92)
Stecco, et al. [12]	Observational study (cohort)	Dissection + Histological	20 female cadavers (≥8 embalmed)	N/A (range 54–72)
Fritsch, et al. [61]	Observational study (cohort)	Dissection + Radiological (MRI) + Histological	6 embalmed female cadavers 41 living females	Cadavers: N/A Living females (N/A): (range 19–43)
El-Sayed, et al. [18]	Observational study (cohort)	Dissection + Radiological (MRI) + Histological	7 embalmed female cadavers and 17 living females	Cadavers: N/A (range 25–50) Living females: 26 (SD 4) (range 20–35)
Nagata, et al. [20]	Observational study (cohort)	Dissection + Histological	20 embalmed female cadavers	82 (range 71–95)
Soga, et al. [62]	Observational study (cohort)	Dissection + Histological	15 embalmed female cadavers	84 (range 66–99)
Betschart, et al. [63]	Observational study (cohort)	Histological	22 embalmed female cadavers	87 (range 74–101)
Garcia-Armengol, et al. [64]	Observational study (cohort)	Dissection	5 embalmed female cadavers	N/A
Kato, et al. [65]	Observational study (cohort)	Dissection + Histological	15 embalmed female cadavers	75 (range 64–90)
Stein and DeLancey [66]	Observational study (cohort)	Dissection	3 female cadavers	N/A (range 28–56)
Brandon, et al. [67]	Observational study (cohort)	Radiological (MRI)	20 living females	N/A (range 23–55)
Larson, et al. [68]	Observational study (cohort)	Radiological (MRI) + Clinical morphometric measure	11 living females	61 (SD 10)
Hirata, et al. [69]	Observational study (case-control)	Histological	17 embalmed female cadavers (and 10 embalmed male cadavers)	N/A (range 56–84) (females and males combined)
Hirata, et al. [70]	Observational study (cohort)	Histological	17 embalmed female cadavers	75 (range 56–91)
Tsai, et al. [71]	Observational study (cohort)	Clinical morphometric measure	200 living females in the first stage of labor	27 (SD 6)
Hinata, et al. [72]	Observational study (cohort)	Dissection + Histological	10 embalmed female cadavers	85 (range 73–100)
Kraima, et al. [21]	Observational study (case-control)	Dissection + Histological	2 embalmed female cadavers (and 2 embalmed male cadavers)	N/A
Santoro, et al. [73]	Observational study (cohort)	Radiological (3D endovaginal ultrasound imaging)	5 fresh female cadavers and 44 living females	Cadavers: N/A Living females: 28 (range 18–34)
Lane, et al. [74]	Observational study (cohort)	Clinical morphometric measure	127 living females in the first stage of labor	24 (SD 5) (range 15–38)
Wu, et al. [75]	Observational study (cohort)	Dissection	5 female cadavers	33 (range 22–59)
Hamner, et al. [22]	Observational study (cohort)	Dissection + Histological	25 female cadavers (≥4 fresh)	76 (range 33–95)
Kochová, et al. [76]	Observational study (cohort)	Histological (including software for quantification)	15 fresh female cadavers	74 (SD 10)
Ghareeb, et al. [77]	Observational study (case-control)	Dissection	5 embalmed female cadavers (and 13 embalmed male cadavers)	74 (SD 7) (range 65–83)
Rodríguez-Abarca, et al. [78]	Observational study (case-control)	Radiological (MRI)	102 living females	41 (SD 15) (range 15–77)
Li, et al. [17]	Observational study (cohort)	Radiological (MRI and 3D reconstruction)	4 female cadavers and 10 living females	Cadavers: N/A (range 22–25) Living females: N/A
Mboua Batoum, et al. [79]	Observational study (cohort)	Clinical morphometric measures	103 living females in the first stage of labor	23 (range 16–40)

MRI: magnetic resonance imagery.

**Table 2 life-11-00900-t002:** Anatomical description based on included studies.

**Structures and Authors**	**Description and Course**	**Length, Mean (SD or Range)**	**Width, Mean (SD or Range)**	**Thickness, Mean (SD or Range)**	**Histological Assessment**
**Perineal membrane (PM)**					
Aronson, et al. [54]	- PM presented distinct anatomical subdivisions.				
DeLancey [55]	- Ran from the ischiopubic rami to the PB. - Orientation of fibers: - Inferior portion of the vagina: fibers lied in a transverse plane spanning the anterior triangle between the ischiopubic rami. - Mid-portion of the vagina: fibers came from parallel fascial sheets in the sagittal plane connecting to the inner surface of the pelvic diaphragm. - A few striated muscle fibers of the superficial transverse muscles lied along the posterior margin of the PM.				
Betschart, et al. [63]					- 80% of the PM consisted of connective tissue. - Smooth and striated muscle were found. - Fat and blood vessels were also identified.
Kato, et al. [65]	- Appearance of a posterolateral fibrous skeleton to reinforce the vagina and pull up its most inferior end. - Distinct layer immediately inferior to the LAM and rhabdosphincter area. - Orientation of fibers: - Oriented sagittally and in continuity with a fascia covering the inferior surface of the LAM. - Extended posteroinferiorly along the lateral vaginal wall. - Anterior portion: Fibers consistently ran transversely. The lateral extension of the PM in the anterior area was thin and interrupted. - Posterior portion: Fibers were more tilted along the superolateral-inferomedial axis. Fibers ran along the vagina and were much more tightly packed than those of the anterior part. PM became tighter and bulkier.			Anterior area (range 3–12 mm)	- Most fibers were elastic. - Collagenous fibers were also found in the perimysium of the embedded striated muscles. - Inferoposterior area: - Fibers presented as a tight meshwork of elastic fibers that were fasciculated to provide a connective tissue plate (5 mm in maximum thickness) that lined and reinforced the lateral vaginal wall - Inferolateral aspect: - Presented as a loose mass of connective tissue, rich in fat in the perineal subcutaneous tissue. - Most of the “compressor-urethrovaginal sphincter complex” muscle fibers were embedded in the PM and interdigitated with composite elastic fibers. - PM provided a thick intramuscular septum or epimysium for these muscles.
Stein, et al. [66]	- Located between the vestibular bulb as well as the clitoral crus and the LAM which were fused to the surface of the PM complex. - LAM fibers inserted directly into the PM. - PM was at the level of the vaginal lumen. - Connective tissue mass increased going from anterior to posterior and from the midline to lateral. - Anterior portion lateral to the urethra - Continuous with the paraurethral (or endopelvic) and paravaginal connective tissues and contained the compressor urethrae and urethrovaginal sphincter muscles of the distal urethra. - Continuous with the superior fascia of the LAM and with the insertion of the TAPF into the inferior pubic ramus. - Posterior portion lateral to the PB - Attached the PB and lateral wall of the vagina to the ischiopubic rami. - Superficial transverse perineal muscle marked the posterior edge of the PM, but was excluded from the PM.				
Brandon, et al. [67]	- Extended from the anterior attachment on the pubic bones to the inferior posterior border on the ischiopubic ramus at the level of the PB. - Anterior portion - Superior surface of the PM was intimately related to the anterior LAM which fused with the paraurethral and paravaginal connective tissues. - Inferior surface of the PM was fused with the superior margins of the vestibular bulb and the clitoral crus with their muscle fibers. - Posterior portion - Separate appearance, single fibrous band extending laterally to the inferior pubic ramus at the level of the vaginal lumen. - Ran in a transverse direction between the midline to the lateral wall. - Superior surface of the PM fused along the fascia of the LAM.				
Hirata, et al. [69]					- Abundant elastic fibers and collagenous fibers found. - Smooth muscle fibers absent or rare; however, smooth muscle fibers of the vaginal wall were attached to the PM with the vestibular bulb.
**Perineal body (PB)**					
DeLancey [55]	- The PB connected the two halves of the PM. - The inferior part of the rectum abutted the PB. - Extended superiorly for approximately 2–3 cm above the hymenal ring. - The lateral margin of the PB contained the termination of the bulbocavernosus muscle.			Thickest in its inferior part and progressively thinner toward its superior margin.	
	- Occupied a space that was surrounded by the vestibular bulb, internal anal sphincter, and levator ani slings. - The authors divided the PB from its lateral extensions.	At the midsagittal section, the antero-posterior length of the PB range was 10–20 mm (dissection).			- Fibromuscular tissue mass between the internal anal sphincter and the vaginal wall. - Smooth muscle fibers were irregularly arrayed and fibers tended to be directed almost horizontally to connect the vagina and the rectum in the superior part near the RVF. - Inferior margin of the PB: loose subcutaneous tissue. - The superior margin of the PB: continued to the RVF and contained blood vessels. - Lateral extension (LEX) of the PB was composed of fibromuscular tissues. - LEX had a higher smooth muscle content than the PB itself. - Striated muscle predominantly located in the LEX than the midline. - Smooth muscles connected to the internal anal sphincter and the vaginal smooth muscles. - LEX occupied a space located between the vestibular bulb (anterior) and the internal anal sphincter (posterior) - LEX was located on the inferior portion of the LAM. - LEX included major anterior insertions of the external anal sphincter. - Bulbospongious extended sometimes posteriorly along the inferior surface of the LEX. - Bartholin’s gland was located along or embedded in the anteroinferior margin of the LEX near the vestibular bulb.
Kato, et al. [65]	- Peripheral extension of the PB provided an inferoposterior attachment for the PM. - The extension of the PB connected tightly to the internal anal sphincter and provided a fibrous skeleton for the posterior vaginal wall.				- Thin smooth muscle fibers were found. - Bulky fibers laterally and anteriorly - Blood vessels also noted.
Larson, et al. [68]	Using MRI: - Pyramidal structure between the vagina and the rectum. - Longitudinal extension of the PB attached from the superficial transverse perineal muscle and external anal sphincter to the fusion of the longitudinal muscle of the rectum with the internal anal sphincter. - Inferior portion of the PB: at the level of the vestibular bulb, the bulbospongiosus inserted into the lateral margin of the PB. The superficial transverse perineal muscle and external anal sphincter traversed the PB. - Midportion of the PB: at the superior end of the superficial transverse perineal muscle, the puboperinealis muscle (component of the pubovisceral muscle) inserted into the lateral margins of the PB. The midportion of the PB also contained the inferior internal anal sphincter. The puboanalis muscle was also visible. Fibers of the pubovisceral muscle extended across the PB. The posterior portion of the external anal sphincter extended to the PB inferiorly. - Superior portion of the PB: puboanalis muscle and the internal anal sphincter extended into the PB’s most superior region at the level of the midurethra. The pubovaginalis muscle was visible as it was fused with the vaginal side wall, sending fibers posteriorly to the PB. The longitudinal muscles of the rectum could be visible in the midline. The puborectal muscle formed a loop behind the rectum but did not contribute fibers to the PB.	3.2 cm (SD 1.3) (clinical measure)			
Tsai, et al. [71]		3.9 (95% CI 3.8–4.0) cm (range 2.2–6.0) (clinical measure)			
Santoro, et al. [73]	- Divided into a superficial level, in contact with the external anal sphincter, the bulbospongiosus and the superficial transverse perineal muscle and a deep level, in contact with puboperinealis and puboanalis muscles	Antero-posterior diameter (depth) 14.49 mm (SD 1.48) as measured with US	Latero-lateral diameter (width) 13.26 mm (SD 1.19) as measured with US	Superior-inferior diameter (height) 7.57 mm (SD 0.45) as measured with US	
Lane, et al. [74]		3.7 cm (SD 0.5) (range 2.3–5.0) (clinical measures)			
Wu, et al. [75]	- Highly irregular fibromuscular structure. - Pyramidal node. - Located between the vagina anteriorly and the anal canal posteriorly, and the anorectal bend superiorly, and the perineal skin inferiorly. - Extensions of the PB formed the tendinous attachments of eight muscles: rectoperineal and deep perineal muscles posteriorly, medial layer of pubovisceral muscle and inferior and superior portions of puborectal muscle laterally, and urethrovaginal sphincter, superficial transverse perineal and bulbospongiosus anteriorly. - Fibers of the bulbospongiosus and superficial transverse perineal muscles were indistinguishable in the midline, and they formed a muscle bundle that attached to the posterior part of the PB. - Expansion of adipose fat tissue in the PB with aging.	Size depended on the fibrous tissue development. Biometry: 2.7 (SD 1.7) mL (n = 4 Asian) and 0.6 mL (n = 1 Caucasian) (dissection).			
Kochova, et al. [76]	- Connected the muscles from the pelvic floor.				- Smooth muscle cells (11%), skeletal muscle fibers (3%), collagen fibers (29%), elastin fibers (7%), adipose cells (27%) and residual tissue (19%) were identified. - No difference in composition (e.g., smooth, striated muscle, collagen, elastin, and adipose tissue) found across 5 different PB areas extending from the lateral sides to midline.
Mboua Batoum, et al. [79]	- Extended from the posterior vaginal fourchette to the center of the anal orifice.	3.21 cm (SD 0.75) (range 1.5–5.5) (clinical measures)			
**Endopelvic fascia**					
DeLancey [16]	- One continuous unit and had distinct areas. - Attached the urinary bladder, uterus, vagina, and rectum to the pelvic walls.				
Aronson, et al. [54]	- Attached to the TAPF, lateral vaginal wall and urethra.				
DeLancey [55]	- Attached the middle portion of the vaginal wall on either side of the rectum. - Most fibers attached to the vaginal wall, with only a few fibers passing from one side to the other.				
Ercoli, et al. [34]	- A variable dense fascial system covering the structures limiting the pelvic cavity: LAM, obturator, ischiococcygeus and piriformis muscles, anterior surfaces of the sacrum and coccyx, and structures contiguous to the pelvic walls such as hypogastric vessels and sacral roots. - Pierced by 4 bilateral foramens for the neurovascular bundles; directed toward the legs and the gluteoperineal region: obturator, superior gluteal, inferior gluteal, and internal pudendal bundles. - Included the superior fascia of pelvic diaphragm (pubovesical ligament), obturator fascia, TAPF, levator ani and rectovaginal fasciae and piriformis fascia. - Four bilateral thickenings: TAPF, TALA, tendinous arch of rectovaginal fascia and pubovesical ligament.				
Hirata, et al. [69]	- Orientation of fibers: - Elastic fibers ran across the sectional plane in most specimens which were perpendicular to most of the LAM fibers. - Elastic fibers seemed to cross the superoinferiorly extending striated muscle fibers.				- Thick and solid monolayer reinforced by abundant elastic fibers running along the anteroposterior axis. - Contained little or no smooth muscles, except for the most posterior areas near the rectum.
**Pubourethral ligament/Pubovesical ligament**					
Milley and Nichols [52]	- Pubourethral ligament suspended the urethra under the arch of the pubis and corresponded to 2 symmetrical dense connective tissue bands firmly attached to the anterior or posterior surface of the pubic bone just lateral to the pubic symphysis on its lower part. - Anterior and posterior bands fanned out as they passed from their superior bony attachment inferiorly to attach to the superolateral aspect of the urethra. - Anterior and posterior bands were joined together under the arch of the pubis in the sagittal plane to form an intermediate ligament which fused with the urethra inferiorly. - Posterior pubourethral ligament: Fibers blended posterolaterally with the TALA. It was a modified reflection of the superior fascia of the endopelvic fascia. - Anterior pubourethral ligament: Fibers were continuous anterosuperiorly with the suspensory ligament of the clitoris and shared the same pubic attachment. It was the superior extension of the inferior fascia of the endopelvic fascia.				
DeLancey [19]	- Pubovesical ligament attached primarily to the TALA rather than the pubic bones and was an extension of the detrusor muscle and its adventia and was attaching to the pubic bone and the TAPF.				- Bands of smooth muscle similar in appearance to, and continuous with, the detrusor muscle were identified and named the pubovesical muscle. - Pubovesical muscle was embedded in connective tissue, and when the muscle fibers and this connective tissue investment were considered together, they were called the pubovesical ligament.
DeLancey [19]	- Pubourethral ligaments supported the urethra. - Ligaments were beside the urethra arising primarily from the vagina and the periurethral tissues to attach laterally to the pelvic wall and were separated from the pubovesical ligament by a para-urethral vascular plexus and lied posterior to the urethra. - They did not attach directly to the pubic bones but inserted into the pelvic fasciae and muscles.				- Dense and fibrous connective tissue was noted.
Aronson, et al. [54]	- Pubourethral ligament was a condensation of tissue from the posterior symphysis pubis to the endopelvic fascia overlaying the anterior vaginal wall. - Could not be reliably subdivided into fascial and muscular components.				
Vazzoler, et al. [24]	- Superior pubourethral ligament: inserted on the posterior surface of the symphysis pubis to the posterior aspect of the superior third of the urethra and bladder neck.	10 mm		3 mm	- This ligament was composed of dense collagen fibers, elastic fibers, and bundles of smooth muscle fibers.
	- Inferior pubourethral ligament: inserted on the inferior border of the symphysis pubis to the distal third of the urethra on its posterolateral surface near the urethral meatus.	N/A (range 2–4) mm			- This ligament had a more loosely woven structure as its composition was mainly of elastic fibers and smooth muscle fibers.
	- Intermediate pubourethral ligament: stretched between the posterior wall of the middle urethra and inferior posterior surface of the symphysis pubis.				- This ligament consisted mainly of loose cellular adipose tissue.
Pit, et al. [59]	- Pubourethral ligament originated on the pelvic bone medial and inferior to the attachment of the TAPF. - Orientation of fibers: - Oriented more toward the midurethra. - Fibers originating from the TAPF were directed more superiorly.				
Ercoli, et al. [34]	- Pubovesical ligament was a prominent bilateral fold of the superior fascia of pelvic diaphragm and was taut between the pubic symphysis and the urinary bladder neck.				
Fritsch, et al. [61]	- Pubovesical ligament was reported as absent.				- The structure that could refer to the pubovesical ligament, located in the retropubic space running from the pubic symphysis to the ventral wall of the urinary bladder neck, did not contained much collagen, but mainly smooth muscle cells. Therefore, the authors recommended that this structure should be referred to as the pubovesical muscle and not a ligamentous structure.
El-Sayed, et al. [18]	- Superior pubourethral ligament: extended from the anterolateral aspect of the superior urethra to the inferior part of the posterior aspect of the pubic bone in an anteroposterior orientation.				- Composed mainly of collagen bundles between which were interspersed smooth muscle fibers.
	- Inferior pubourethral ligament: extended from the inferior third of the urethra and extended to the inferior border of the pubic bone on its anterior surface in an anteroposterior orientation. - Intermediate pubourethral ligament: extended from the middle third of the urethra to the inferior border of the pubic bone in an anteroposterior orientation.				- Consisted mainly of dense collagen bundles that were oriented in a variety of directions. - Composed of collagen bundles that were isolated within a connective tissue stroma with a few capillaries and scattered muscle fibers.
**Tendinous arch of pelvic fascia (TAPF)**					
Hamner, et al. [22]					- Fat tissue, blood vessels and nerves between the urinary bladder and the anterior vaginal wall. - No structure consistent with a discrete fascial layer was noted between the bladder\urethra and the anterior vagina. - Condensations of connective tissue between the urethra and the pubic bone at the TAPF were noted on gross examination but no discrete tissue consistent with pubourethral ligaments was identified.
Li, et al. [17]	- Pubovesical ligament was located between the pubis and the main part of the urethral sphincter and was directly connected to the pubovesical muscle	12.3 mm (SD 5.0)			
Occelli, et al. [58]	- Fibrous thickening of the pelvic fascia that partially constituted Roggie’s star. - Inserted on the posterior inferior aspect of the pubic symphysis at the level of the pubovesical ligament. - Posterior third of the TAPF fused with the posterior third of the TALA and related laterally to the obturator internus muscle and medially to the pelvic peritoneum. - Lower portion of the TAPF came from the fascia of the LAM, the superior posterior third of fibers from the TALA. - Superior portion and lateral part of the TAPF came from the fascia of the obturator internus muscle. - Inferior and lateral portion of the TAPF came from the superior fascia of the pelvic diaphragm. - Superior tendinous thickening of Roggie’s star was the extension of the TAPF. - This thickening ran upward and posteriorly and became impossible to distinguish amidst the fasciae of the piriformis and obturator internus muscle.	10 mm			
Pit, et al. [59]	- Condensation of the endopelvic fascia and presented as a white line within the fascia covering the pubococcygeus and iliococcygeus muscles. - Extended from the inferior inner surface of the body of the pubic bone to the most medial tip of the ischial spine and could be merged with the TALA. - The first anterior cm of the TAPF had a clear lateral fixation either to the lateral part of the LAM, to the fascia covering the obturator internus muscle, or to the obturator membrane. - Derived from a long flat fibrous attachment to the posterolateral aspect of the urethra. - Formed a strong fibrous plate in the sagittal plane that gradually decreased in size as it continued posteriorly toward the ischial spine.				
Ersoy, et al. [23]	- Formed from the fan shaped attachment of the cardinal ligament to the pelvic fascia, covering the inner surface of the LAM.	8.1 cm (range 7.5–9.0)			
Albright, et al. [60]	- Extended from the ischial spine through the RVF and TALA to the body of the pubic bone. - Condensations of the TALA and the RVF attached at the same point in the TAPF, and fibers of the RVF could attached to the lateral border of the TAPF.	8.99 cm (SD 0.70) (range 7.0–10.5); variability between right and left TAPF (range 0–0.75 cm)			
Ercoli, et al. [34]	- Formed by the line of fusion of the pubocervical fascia with the endopelvic fascia covering the obturator muscle. - Extended from the ischial spine to the inferior surface of the pubic bone and symphysis.				
Brandon, et al. [67]	- Originated from the fibrous complex of the symphysis just superior to the paraurethral connective tissues (inferior surface of the pubic symphysis). - Continuous with the fascia of the anterior LAM which formed the anterior portion of the PM. - Fused with the paraurethral and paravaginal connective tissue which contained the compressor urethrae and the urethrovaginal sphincter muscles of the distal urethra.				
Hirata, et al. [69]					- Mesh-like architecture, largely composed of collagenous fibers that ran along the mediolateral axis - Abundant elastic fibers were also found, usually oriented in a mediolateral direction. - Smooth muscle fibers were densely packed near the rectovaginal fascia and scattered near the pubocervical fascia.
Li, et al. [17]	- Fibrous structure with a curved line shape. - Originated at the medial inferior part of the posterior surface of the pubis and lateral to the pubic symphysis and inserted at the ischial spine. - Located beneath the TALA. - Anterior portion run onto the LAM and connected the lateral vesical ligament medially; its posterior part ran upward and onto the internal obturator muscle.	81.1 mm (SD 2.4) and divided in 2 parts at its conjoining point with the TALA; anterior portion 53.8 mm (SD 20.6); posterior portion 31.6 mm (SD 18.6)	Anterior portion 4.0 mm (SD 0.6); posterior portion 10.8 mm (SD 3.1)	The posterior portion was much thicker than the anterior portion.	
**Tendinous arch of levator ani (TALA)**					
Pit, et al. [59]	- Well-developed, slightly arched white line and was not part of the endopelvic fascia. - Located just anterior to the obturator canal. - Extended from the inner surface of the superior ramus of the pubic bone to the ischial spine which was clearly distinctive from the TAPF as it was more lateral than the TAPF. - Formed the superior border of the connective tissue plate from whose the posterior part of the pubococcygeus muscle and iliococcygeus muscle originated.				
Ersoy, et al. [23]	- TALA = origin of the LAM. - Thickening of the obturator fascia or a separate entity like a bridge holding onto the pubis anteriorly and the sacrum posteriorly, like a fibrous structure. - Originated from the superior pubic ramus or the inferior border of the obturator internus.				
Ercoli, et al. [34]	- Merged with the TAPF from the ischial spine to about the middle portion of the TAPF at which point TALA diverged to rejoin the inferior pubic ramus, allowing insertion of the pubococcygeus, puborectal, and pubovaginal portions of the LAM.				
Li, et al. [17]	- Constituted a tendinous arch in the pelvic wall. - Originated from the pubis anteriorly and ended at the conjoining point with the TAPF posteriorly.	48.0 mm (SD 17.4)		TALA thinner than the TAPF.	
**Pubocervical fascia**					
DeLancey [16]	- Connected the anterior wall to the TAPF and formed a supportive layer that lied under the urinary bladder and in continuity with the RVF.				
Ercoli, et al. [34]	- Originated in the TAPF and attached the lateral portions of the middle third of the vagina to the pelvic walls to determine the H-shaped configuration of the vagina at this level. - Fused with the anterior surfaces of the vaginal fascia/vaginal wall.			Became progressively thinner, from the perineum toward the vaginal apex.	
Hinata, et al. [72]	- Pubocervical fascia was identifiable as a rough fibrous network of thick fiber bundles in the lateral area. - Pubocervical fascia provided a boundary for the paracolpium facing the thick subperitoneal fatty tissue. - Pubocervical fascia was often independently connected to the superior fascia of the LAM. - Pubocervical fascia sandwiched the pelvic autonomic nerve plexus.				- No smooth muscle was found.
**Rectovaginal fascia (RVF)**					
DeLancey [16]	- Corresponded to the superior fascia of the LAM that attached to the posterior wall of the vagina and was in continuity with the pubocervical fascia.				
Leffler, et al. [57]	- Extended from the pubic bone toward the TAPF which converged approximately midway between the pubis and the ischial spine to form a Y configuration. - Fused with the aponeurosis of the LAM that began at the PB and converged at the TAPF. - The point of convergence occurred at an average of 4.80 cm from the ischial spine, 3.75 cm from the pubic symphysis, and 4.15 cm from the posterior fourchette.				
Ercoli, et al. [34]	- Attached to the lateral portions of the middle third of the vagina to the pelvic walls to determine the H-shaped configuration of the vagina at this level. - Fused with the posterior surfaces of the vaginal fascia/vaginal wall to form a single structure.			Progressively thinner, from the perineum toward the vaginal apex.	
Stecco, et al. [12]	- Fibrous tissue between the rectum and the posterior vaginal wall. - In an oblique coronal plane, superiorly attached to the peritoneum of the recto-uterine pouch and inferiorly to the PB. - Extended from the anterior aspect of the anorectal junction and the posterior aspect of the vaginal wall to the PB. - Fused with the fascia of the LAM.			Superior part of the median third of the vagina 1.75 mm (SD 0.75). Superior part of the lower third in the midline 1.70 mm (SD 0.88). Lateral portions of RVF 2.67 mm (SD 1.08), 2.64 mm (SD 1.12). Inferior median third at the inferior part in the midline 0.20 mm (SD 0.11), and laterally 0.17 mm (SD 0.07).	- Collagen and elastic fibers. - Smooth muscle cells, with variable numbers of small blood vessels were also identified. - There was a double fibro-connective lamina close to the posterior vaginal wall, and RVF was considered as the outer lamina which connected laterally with the septa of the subperitoneal fibro-adipose connective tissue. - Composition of the RVF was different in the inferior-superior axis. - Superior slices levels II and III: fibro-adipose tissue (thicker than inferior levels); thin layer of connective tissue surrounding thin transverse smooth muscle bundles; thin laminae were interconnected forming a network; connective laminae showed a vertical course; elastic fibers were poorly represented. - Inferior slices levels II: fibro-adipose tissue; thin layer of connective tissue meshed with smooth muscle bundles and was more closely packed; adipose tissue was poorly represented at midline; elastic fibers were identifiable in the lateral parts of the RVF and showed a well-developed muscular tunica; connective laminae were interconnected forming a dense network. - Inferior slices level III: the two laminae were not identified; smooth muscle bundles were intermingled with fibrous tissue, showing a sagittal course. Connective tissue showed a vertical course. Adipose tissue was lacking.
Nagata, et al. [20]	- Tight, thick, straight, and evident in the inferior half and loose and thin in the superior half.	35–60 mm (as defined between the peritoneal reflection at the Douglas’s pouch and the superior end of the internal anal sphincter).		Ranged from 0.1 mm to 0.3 mm, depending on the sites or levels of the vagina.	- Posterior vaginal wall: an elastic fiber-rich plate (EFRP) was sometimes evident. - EFRP was composed of two layers, because of the elastic fiber layers, sandwiched veins. - No or few smooth muscles were found in the EFRP. - EFRP became thin and interrupted by veins in the upper half. - EFRP extended between the bilateral paracolpiums. - Parts of the lateral end of the EFRP appeared to attach the parietal fascia covering the LAM. - EFRP was considered as the RVF because of the specific morphology.
Hinata, et al. [72]	- Fibrous boundary facing the mesorectal loose tissue in the medial area near the midsagittal line and was sometimes fragmented or unclear in the lateral area. - Independently connected to the superior fascia of the LAM. - RVF sandwiched the pelvic autonomic nerve plexus.				- Collagenous and elastic fibers, no smooth muscle was found.
Kraima, et al. [21]	- Situated between the cervix uteri, the vagina, and the rectum. - Extended from the recto-uterine pouch to its termination in the PB. - Appeared fan-shaped at the level of the cervix uteri and upper vagina, - Posterior portion at midline fused with the mesorectal fascia in a lateral direction and the Denonviliers’ fascia curved along it.				- Collagen and smooth muscle fibers were found. - The posterior portion to the vagina contained more elastin.
Ghareeb, et al. [77]	- Attached to the deepest part of the peritoneal reflection and to the PB. - Completely separated from the proper fascia of the rectum posteriorly and the posterior wall of the vagina anteriorly. - Above the S4 level, merged posterolaterally to attach to the rectosacral fascia. - Below the S4 level, the lateral attachment could be traced with the presacral fascia along its origin from the TALA.				
Rodriguez-Abaraca, et al. [78]		73.2 mm (SD 15.3), (mid-sagittal plane by using the posterior fornix and the end of the recto-uterine pouch to the PB)		Superior third 2.8 mm (SD 1.7), middle third 2.2 mm (SD 1.2), inferior third 2.5 mm (SD 1.3)	
**Tendinous arch of rectovaginal fascia**					
Ercoli, et al. [34]	- Taut band between the PB and the middle third of the vagina at which point it merged with the TAPF. - Formed by the line of fusion of the rectovaginal fascia with the endopelvic fascia covered the LAM.				
**Rectovaginal ligament**					
Ercoli, et al. [34]	- Originated from the dorsolateral portions of pelvic walls situated between S2 and S4 sacral foramens and the ischial spines, which constituted a center of convergence of the fibers of the endopelvic fascia and the visceral fascia, and converged toward the proximal and middle vaginal thirds.				
**Rectosacral fascia/** **Inferior fascia of the pelvic diaphragm**					
Ercoli, et al. [34]	- Fused anteriorly with the presacral fascia and laterally with the rectal stalks.				
Garcia-Armengol, et al. [64]	- Originated from the presacral parietal fascia, passing inferiorly to join the mesorectal visceral fascia 3–5 cm above the anorectal junction. - Fibers originated variably from S3 or S4 - Divided the retrorectal space into two portions: inferior and superior.				
	- The inferior fascia of the pelvic diaphragm was the floor or the inferior limit of the inferior portion, and of the retrorectal space overall.				
**Paracolpium**					
DeLancey [16]	- Attached the vagina to the pelvic walls and was located at level I and II. - Continuous with the cardinal and uterosacral ligaments. - Originated from the greater sciatic foramen over the piriformis muscles, from the pelvic bones in the region of the sacroiliac joint, and from the lateral sacrum. - Orientation of fibers: - Primarily vertical and some posteriorly from the vagina toward the sacrum in a more horizontal direction.				
Hinata, et al. [72]					- Paracolpium contained elastic fibers, blood vessels and nerves.

TAPF: tendinous arch of pelvic fascia, TALA: tendinous arch of levator ani, LAM: levator ani muscles, PB: perineal body, PM: perineal membrane, RVF: rectovaginal fascia.

**Table 3 life-11-00900-t003:** Risk of bias assessment.

Authors	Domain 1 Objectives and Subject Characteristics	Domain 2 Study Design	Domain 3 Characterization of Methods	Domain 4 Descriptive Anatomy	Domain 5 Results Reporting	Overall Risk of Bias
Milley and Nichol [52]	Unclear	High	High	Unclear	High	High
DeLancey [19]	Unclear	Unclear	High	High	High	High
DeLancey and Starr [53]	Unclear	Unclear	Unclear	Unclear	High	High
DeLancey [16]	Unclear	Unclear	Unclear	Unclear	High	High
Aronson, et al. [54]	Unclear	Low	Low	Unclear	Unclear	High
DeLancey [55]	Unclear	Unclear	Low	Low	Unclear	High
Mauroy, et al. [56]	High	Low	High	High	High	High
Leffler, et al. [57]	Unclear	Low	Unclear	Low	Low	High
Occelli, et al. [58]	High	Unclear	Unclear	Unclear	Unclear	High
Vazzoler, et al. [24]	Unclear	Unclear	Unclear	Low	Low	High
Pit, et al. [59]	Unclear	Low	Low	Low	Low	Unclear
Ersoy, et al. [23]	Unclear	Unclear	Low	Low	High	High
Albright, et al. [60]	Low	Low	Low	Low	Low	Low
Ercoli, et al. [34]	Low	Low	Low	Unclear	Unclear	High
Stecco, et al. [12]	Low	Low	Low	Unclear	Low	Unclear
Fritsch, et al. [61]	Unclear	Low	Unclear	Unclear	Unclear	High
El-Sayed, et al. [18]	Low	Low	Low	Low	Low	Low
Nagata, et al. [20]	Low	Low	Low	Low	Low	Low
Soga, et al. [62]	Low	Low	Low	Low	Low	Low
Betschart, et al. [63]	Low	Low	Low	Low	Low	Low
Garcia-Armengol, et al. [64]	Unclear	Low	Low	Low	Unclear	High
Kato, et al. [65]	Low	Low	Low	Low	Low	Low
Stein and DeLancey [66]	Unclear	Low	Unclear	Low	Low	High
Brandon, et al. [67]	Unclear	Low	Low	Unclear	High	High
Larson, et al. [68]	Low	Low	Low	Low	Low	Low
Hirata, et al. [69]	Low	Low	Low	Low	Low	Low
Hirata, et al. [70]	Low	Low	Low	Low	Low	Low
Tsai, et al. [71]	Low	Low	Low	Unclear	Unclear	High
Hinata, et al. [72]	Low	Low	Low	Low	Low	Low
Kraima, et al. [21]	Unclear	Low	Low	Low	Low	Unclear
Santoro, et al. [73]	Low	Low	Low	Unclear	Unclear	High
Lane, et al. [74]	Low	Low	Low	Low	Low	Low
Wu, et al. [75]	Unclear	Low	Unclear	Unclear	Unclear	High
Hamner, et al. [22]	Low	Low	Low	Low	Low	Low
Kochova, et al. [76]	Low	Low	Low	Low	Low	Low
Ghareeb, et al. [77]	Low	Low	Low	Low	Low	Low
Rodriguez-Abarca, et al. [78]	Low	Low	Low	Low	Low	Low
Li, et al. [17]	Unclear	Unclear	Unclear	Unclear	Unclear	High
Mboua Batoum, et al. [79]	Low	Low	Low	Low	Low	Low

## Data Availability

Not applicable.

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
