# Peer review of "The Female Pelvic Floor Fascia Anatomy: A Systematic Search and Review"

_life, 2021, doi:10.3390/life11090900_

Round 1

Reviewer 1 Report

Review of the systematic search and review

“The female pelvic floor fascia anatomy: a systematic search and research”

The presented study is a large review of literature on the so-called pelvic fascia.

The review is important and necessary, but in some aspects it lacks a clear morphological basic. In the results part as well as in the discussion part different structures are listed. It lacks a definition of ligaments and fascia in the introduction part. Are the pelvic structures studied are ligaments and fascia as defined (please use actual text books of anatomy or histology?). Please clarify.

Furthermore, terms are used and listed that are not included in the Terminologia Anatomica such as Waldeyer´s fascia, Colles and Galaudet´s fasciae… Please use the Terminologia throughout the paper.

The completeness of the summarized literature is debatable, some important papers are not cited nor included (for example)

Introduction part: line 46: Höckel et al. 2005 Lancet Oncol; line 52: Aigner et al. 2004 Dis Colon Rectum and Fritsch et al. 2006  Neurourol Urodyn and Ludwikowski et al. 2002 JPS; line 76: Fritsch and Hötzinger 1995 Clin Anat and Fröhlich et al. 1997 Clin Anat and Fritsch et al. 2004 Adv Anat Cell Biol Embryol and Fritsch et al. 2012 Worl J Urol

Discussion part: The content of the mentioned papers must be included in the discussion part.   

Author Response

Dear reviewer,

We would like to thank you for the interest and taking time off your busy schedules to review our manuscript. Your highlights and suggestions were immensely helpful and significantly improved our contribution.

All revisions made to the manuscript were approved by all authors and are highlighted as requested by the editor. Below we include a response to each of your comments indicating how we have addressed them.

We hope the revised manuscript satisfies all of your queries. Otherwise, we stand ready to consider further revisions and we thank you again for your time and interest into our research.

Sincerely,

Main author of the manuscript LIFE-1336758

Comment 1: The review is important and necessary, but in some aspects it lacks a clear morphological basic. In the results part as well as in the discussion part different structures are listed. It lacks a definition of ligaments and fascia in the introduction part. Are the pelvic structures studied are ligaments and fascia as defined (please use actual text books of anatomy or histology?). Please clarify.

Answer 1: The reviewer raised an important issue.  The goal of our review was to describe the micro- and macro-anatomy of the pelvic fasciae. However, as there is a critical lack of standardisation for discriminating the different types of connective tissues (e.g., fasciae, ligaments, membrane), we opted for a more inclusive approach.  We therefore added in the introduction and methodolody section specifications to clarify the broad scope of our review in terms of the connective tissues included.

Comment 2: Furthermore, terms are used and listed that are not included in the Terminologia Anatomica such as Waldeyer´s fascia, Colles and Galaudet´s fasciae… Please use the Terminologia throughout the paper.

Comment 3:

The completeness of the summarized literature is debatable, some important papers are not cited nor included (for example).

Answer 3: We would like to thank the reviewer for proposing papers to ensure the completeness of our review.  Despite an extensive selection of search terms, the papers suggested were not captured by our search. We carefully examined each manuscript suggested for inclusion in our review.

Höckel M, Horn LC, Fritsch H. Association between the mesenchymal compartment of uterovaginal organogenesis and local tumour spread in stage IB-IIB cervical carcinoma: a prospective study. Lancet Oncol. 2005 Oct;6(10):751-6. doi: 10.1016/S1470-2045(05)70324-7. Epub 2005 Sep 8. PMID: 16198980.

This study was excluded because it included patients with carcinoma. Pathological population was an exclusion criterion in order to capture normal anatomy.

Aigner F, Zbar AP, Ludwikowski B, Kreczy A, Kovacs P, Fritsch H. The rectogenital septum: morphology, function, and clinical relevance. Dis Colon Rectum. 2004 Feb;47(2):131-40. doi: 10.1007/s10350-003-0031-8. PMID: 15043282.  The rectogenital septum: morphology, function, and clinical relevance - PubMed (nih.gov)

We excluded studies conducted in foetus as we focused on the anatomy of adult women.

Fritsch H, Pinggera GM, Lienemann A, Mitterberger M, Bartsch G, Strasser H. What are the supportive structures of the female urethra? Neurourol Urodyn. 2006;25(2):128-34. doi: 10.1002/nau.20133. PMID: 16353239. What are the supportive structures of the female urethra? - PubMed (nih.gov)

We would like to thank the reviewer for suggesting this reference. This study has been added to our review and was taken into consideration in the results section concerning the supporting structures of the urethra 3.4.4. as well as in the discussion section.

Ludwikowski B, Hayward IO, Fritsch H. Rectovaginal fascia: An important structure in pelvic visceral surgery? About its development, structure, and function. J Pediatr Surg. 2002 Apr;37(4):634-8. doi: 10.1053/jpsu.2002.31624. PMID: 11912525. Rectovaginal fascia: An important structure in pelvic visceral surgery? About its development, structure, and function - PubMed (nih.gov)

This study was excluded as it encompassed a fœtal population.

Fritsch H, Hötzinger H. Tomographical anatomy of the pelvis, visceral pelvic connective tissue, and its compartments. Clin Anat. 1995;8(1):17-24. doi: 10.1002/ca.980080103. PMID: 7697508. Tomographical anatomy of the pelvis, visceral pelvic connective tissue, and its compartments - PubMed (nih.gov)

This study was excluded from our review because it did not describe the structures comprised of our pre-determined area of interest.

Fröhlich B, Hötzinger H, Fritsch H. Tomographical anatomy of the pelvis, pelvic floor, and related structures. Clin Anat. 1997;10(4):223-30. doi: 10.1002/(SICI)1098-2353(1997)10:4<223::AID-CA1>3.0.CO;2-T. PMID: 9213037. Tomographical anatomy of the pelvis, pelvic floor, and related structures - PubMed (nih.gov)

This study was excluded because it described the anatomy of men, women and fetuses without providing a separate description for women.

Fritsch H, Lienemann A, Brenner E, Ludwikowski B. Clinical anatomy of the pelvic floor. Adv Anat Embryol Cell Biol. 2004;175:III-IX, 1-64. doi: 10.1007/978-3-642-18548-9. PMID: 15152384. Clinical anatomy of the pelvic floor - PubMed (nih.gov)

This reference was excluded because it is a clinical commentary.

Fritsch H, Zwierzina M, Riss P. Accuracy of concepts in female pelvic floor anatomy: facts and myths! World J Urol. 2012 Aug;30(4):429-35. doi: 10.1007/s00345-011-0777-x. Epub 2011 Oct 15. PMID: 22002833. Accuracy of concepts in female pelvic floor anatomy: facts and myths! - PubMed (nih.gov)

This reference was excluded because it is a scoping review, but it has been added and considered in the discussion section.

Thank you again for your precious time 

Reviewer 2 Report

The article is a very important contribution for larger spread of knowledge concerning the fascial structures of the female pelvis. I definitively recommend its publication after following changes and amendments are performed:

•    Inclusion criteria – why German as a language of anatomy was excluded? These articles should be added.
•    The data contained in classical anatomical works published before 1950 were neglected. Why? They should be added as well.
•    Generally, the TA term „pelvic fascia“, especially in the heading 3.4.3., consisting of parietal and visceral pelvic fascia. Later, terms used in the articles can be stated in brackets to distinguish to proper terminology and the „terminology“ used by individual authors.
•    Minor remarks
120 – change „Colle’s fascia“ to „Colles’ fascia“
359 – change „coccygeus“ to „ischiococcygeus“
360 – change „pelvis diaphragm“ to „pelvic diaphragm“
362 – change „bladder“ to „urinary bladder“ here and further
363 – change „regions“ to „areas“ (as region is a specific anatomical term for limited and defined areas in nomenclature)
366 – what is „vaginal septum“?
378 – „into two distinct portions“ – right and left or? Please, complete. Here and further in relation to three portions.
408 – change „an histological approach“ to „a histological approach“
420 – „superior aspect of the pubic rami“ – do you mean bilaterally or do you mean both the superior and inferior ramus? Please, elucidate.
3.4.7 – I miss here information about the relationship of the pubocervical fascia and the urinary bladder which is located between the pubic symphysis and the vagina/cervix of uterus. Please, complete.
441 – change „identified on the lateral part“ to „identified within the lateral part“
454 – prefer usage of systemic terms to eponyms and change „pouch of Douglas“ to „recto-uterine pouch“ here and further throughout the text
455 – „superior end of the anal sphincter” – which sphincter? Please, complete.
476 – change „endopelvic fascia that covered the LAM“ to „superior facia of pelvic diaphragm“
3.4.11 Rectosacral fascia and Waldeyer’s fascia – do you consider them as synonyms? If yes, then delete „and“ and replace the eponym within the paragraph. If not, explain the difference.
489 – change „presacral parietal fascia“ to „presacral fascia (part of the parietal pelvic fascia)
489 – „mesorectal visceral fascia“ – what is it?
491 – change „retroectal space“ to „retrorectal space“
551 – replace „Gallaudet’s fascia“ with proper anatomical term
672+673 – change „centraum tendineum perinei“ to „centrum tendineum perinei“
735 – change „region around the vesical neck“ to „area around the neck of the urinary bladder“
764 – change „anatomic“ to „anatomical“
797 – change „arcustendineus of the pelvic fascia“ to „TAPF“
800 – change „between the anterior vagina 800 and the bladder“ to „between the anterior aspect of the vagina and the posterior aspect of the urinary bladder“
807 – prefer „rectovesical sspetum“ to „Denonvillier’s fascia“
859 – change „urether“ to „ureter“

Author Response

Dear reviewer,

We would like to thank you for the interest and taking time off your busy schedules to review our manuscript. Your highlights and suggestions were immensely helpful and significantly improved our contribution.

All revisions made to the manuscript were approved by all authors and are highlighted as requested by the editor. Below we include a response to each of your comments indicating how we have addressed them.

We hope the revised manuscript satisfies all of your queries. Otherwise, we stand ready to consider further revisions and we thank you again for your time and interest into our research.

Sincerely,

Main author of the manuscript LIFE-1336758

Comment 1 : Inclusion criteria – why German as a language of anatomy was excluded? These articles should be added.

Answer 1: Unfortunately, we could not include German in our inclusion criteria as no member of our team speaks German.

Comment 2: The data contained in classical anatomical works published before 1950 were neglected. Why? They should be added as well.

Answer 2: The date was not an exclusion criteria. Our search was conducted in various databases from inception, but no study conducted before 1950 met our inclusion criteria.

Comment 3: Generally, the TA term “pelvic fascia”, especially in the heading 3.4.3., consisting of parietal and visceral pelvic fascia. Later, terms used in the articles can be stated in brackets to distinguish to proper terminology and the terminology used by individual authors.

Answer 3: We modified the manuscript accordingly.

Comment 4:  Minor remarks

Answer 4: Modifications have been made accordingly.

120 – change „Colle’s fascia“ to „Colles’ fascia“ (Done)

359 – change „coccygeus“ to „ischiococcygeus“ (Done)

360 – change „pelvis diaphragm“ to „pelvic diaphragm“ (Done)

362 – change „bladder“ to „urinary bladder“ here and further (changed everywhere)

363 – change „regions“ to „areas“ (as region is a specific anatomical term for limited and defined areas in nomenclature) (Done)

366 – what is „vaginal septum“? It was a mistake, it should have read: Rectovaginal fascia, thank you for bringing that up.

378 – „into two distinct portions“ – right and left or? Please, complete. Here and further in relation to three portions. This was clarified.

408 – change „an histological approach“ to „a histological approach“ (Done)

420 – „superior aspect of the pubic rami“ – do you mean bilaterally or do you mean both the superior and inferior ramus? Please, elucidate. It has been specified, bilaterally.

3.4.7 – I miss here information about the relationship of the pubocervical fascia and the urinary bladder which is located between the pubic symphysis and the vagina/cervix of uterus. Please, complete. See section 3.4.7, we re-arranged the information and added some details to make it clearer.

441 – change „identified on the lateral part“ to „identified within the lateral part“ (Done)

454 – prefer usage of systemic terms to eponyms and change „pouch of Douglas“ to „recto-uterine pouch“ here and further throughout the text (Done all over the manuscript)

455 – „superior end of the anal sphincter” – which sphincter? Please, complete.

Added the external sphincter

476 – change „endopelvic fascia that covered the LAM“ to „superior facia of pelvic diaphragm“

(Done)

3.4.11 Rectosacral fascia and Waldeyer’s fascia – do you consider them as synonyms? If yes, then delete „and“ and replace the eponym within the paragraph. If not, explain the difference. We have added some information to make it understandable for the reader and specified that some authors considered it was 2 distinct structures (see 3.4.11)

489 – change „presacral parietal fascia“ to „presacral fascia (part of the parietal pelvic fascia) (Done)

489 – „mesorectal visceral fascia“ – what is it? Thank you for highlighting that matter, we have changed it to the fascia propria of the rectum.

491 – change „retroectal space“ to „retrorectal space“ (Done)

551 – replace „Gallaudet’s fascia“ with proper anatomical term. Since this was a comment from another reviewer, the term has been changed to “perineal fascia”

672+673 – change „centraum tendineum perinei“ to „centrum tendineum perinei“ (Done)

735 – change „region around the vesical neck“ to „area around the neck of the urinary bladder“ (Done)

764 – change „anatomic“ to „anatomical“ (Done)

797 – change „arcustendineus of the pelvic fascia“ to „TAPF“ (Done)

800 – change „between the anterior vagina 800 and the bladder“ to „between the anterior aspect of the vagina and the posterior aspect of the urinary bladder“ (Done)

807 – prefer „rectovesical septum“ to „Denonvillier’s fascia“ (Done)

859 – change „urether“ to „ureter“ (Done)

Thank you again for your precious time

Reviewer 3 Report

The Authors provided a systematic search and review on the "female pelvic floor fascia anatomy". This is still a debated argument and terminology is often different among studies published in the literature. The Authors used a rigorous methodology search and analysed all the relevant studies on this topic. The results provided help in understing the complex anatomy of the pelvic floor and make it clearer.

The only limitation is the High risk of bias of many article assessed, but this was carefuly considered by the Authors in drawing their conclusions

Author Response

Dear reviewer,

We would like to thank you for the interest and taking time off your busy schedules to review our manuscript. Your highlights and suggestions were immensely helpful and significantly improved our contribution.

All revisions made to the manuscript were approved by all authors and are highlighted as requested by the editor. Below we include a response to each of your comments indicating how we have addressed them.

We hope the revised manuscript satisfies all of your queries. Otherwise, we stand ready to consider further revisions and we thank you again for your time and interest into our research.

Sincerely,

Main author of the manuscript LIFE-1336758

Reviewer 3 :

The Authors provided a systematic search and review on the "female pelvic floor fascia anatomy". This is still a debated argument and terminology is often different among studies published in the literature. The Authors used a rigorous methodology search and analysed all the relevant studies on this topic. The results provided help in understing the complex anatomy of the pelvic floor and make it clearer.

The only limitation is the High risk of bias of many articles assessed, but this was carefuly considered by the Authors in drawing their conclusions.

Answer to reviewer 3:

We would like to thank you for your time and the appreciation of our review. Your comments and point of view are very important to us.

Round 2

Reviewer 2 Report

For any future work find a member speaking German as German classical works form an important part of scientific contributions to all fields of anatomy.